# An Application of Machine-Learning Model for Analyzing the Impact of Land-Use Change on Surface Water Resources in Gauteng Province, South Africa

Eskinder Gidey [1,2,]* and Paidamwoyo Mhangara [1]

1  School of Geography, Archaeology and Environmental Studies, Faculty of Science,
   University of Witwatersrand, Johannesburg P.O. Box 2050, Gauteng, South Africa; paida.mhangara@wits.ac.za
2  Department of Land Resources Management and Environmental Protection, College of Dryland Agriculture
   and Natural Resources, Mekelle University, Mekelle P.O. Box 231, Ethiopia
*  Correspondence: eskinder.gidey@mu.edu.et; Tel.: +27-63-558-0069

**Abstract:** The change in land-use diversity is attributed to the anthropogenic factors sustaining life. The surface water bodies and other crucial natural resources in the study area are being depleted at an alarming rate. This study explored the implications of the changing land-use diversity on surface water resources by using a random forest (RF) classifier machine-learning algorithm and remote-sensing models in Gauteng Province, South Africa. Landsat datasets from 1993 to 2022 were used and processed in the Google Earth Engine (GEE) platform, using the RF classifier. The results indicate nine land-use diversity classes having increased and decreased tendencies, with high F-score values ranging from 72.3% to 100%. In GP, the spatial coverage of BL has shrunk by 100.4 km$^2$ every year over the past three decades. Similarly, BuA exhibits an annual decreasing rate of 42.4 km$^2$ due to the effect of dense vegetation coverage within the same land use type. Meanwhile, water bodies, marine quarries, arable lands, grasslands, shrublands, dense forests, and wetlands were expanded annually by 1.3, 2.3, 2.9, 5.6, 11.2, 29.6, and 89.5 km$^2$, respectively. The surface water content level of the study area has been poor throughout the study years. The MNDWI and NDWI values have a stronger Pearson correlation at a radius of 5 km (r = 0.60, $p$ = 0.000, $n$ = 87,260) than at 10 and 15 km. This research is essential to improve current land-use planning and surface water management techniques to reduce the environmental impacts of land-use change.

**Keywords:** land-use diversity; GEE; RF; RS; GIS; Gauteng Province; South Africa

## 1. Introduction

Since the 1960s, the biophysical features of the earth's surface have changed by an average value of 720,000 kilometers square (km$^2$) globally each year [1]. As a result, potential natural resources (e.g., surface water bodies) have been significantly diminishing occasionally. Consequently, knowledge of the real-time land-surface processes, diversity, and change requires regular quantification, monitoring, modeling, analysis, and mapping of the spatial and temporal dynamics of the land-use and land-cover change (LULCC). LULCC describes the transition or alteration of one land-use and land-cover (e.g., water bodies) type into another (e.g., wetlands). Contrarily, land-use diversity is defined as the range of land-use types within a given area [2]. Studies indicate that humans are the main triggering factors of LULCC and have altered about three-quarters of the earth's land surface in the past millennium. For instance, Winkler et al. [1] reported the loss of global forest area by 0.8 million km$^2$ and increases in global agriculture (i.e., cropland and pasture/rangeland) by 1.0 and 0.9 million km$^2$, respectively. The same authors have pointed out that the loss is highly confined in developing countries because humans have produced excessive amounts of charcoal and extracted firewood, resulting in high deforestation rates and temperature change. Therefore, understanding the status of natural resources (e.g.,

water resources; cold and warm grasses; and deciduous, coniferous, and tropical trees) is critical [1,3] for natural-resource conservation. One of the most important reasons is that less than 25% of earth's ice-free territory shows signs of human habitation, and more than 75% of the land use has been significantly altered [4] due to the high demand for housing construction even in the protected parks or forest areas (e.g., Zimbabwe National Park) [5].

The rapid expansion of human settlement has resulted in the loss of potential cropland and water resources [6]. Additionally, the spatial extent of forest land has decreased by 17% [7]. This challenge is quite common in most parts of African countries, e.g., Hugumburda State Forest in Tigray, Ethiopia [8]. As a result, the dynamics of LULCC causes severe vegetation degradation, resulting in the loss of endemic biodiversity, air quality, food supply, and plant species. It also decreases ecological services, intensifies extreme climate, and affects the hydrological cycle, land–air interaction, and ecosystem health [9]. This may continue as one of the regional and worldwide environmental issues that have triggered social, economic, and political crises because of the dependency on natural resources [9,10]. For instance, the decline of forest coverage because of LULCC may reduce the capability of water retention and recharging.

Understanding the spatial patterns and processes will make it easier to predict where and how quickly the land will change and improve the existing land-use planning [11]. Knowledge of land-use diversity is therefore essential for both management and conservation practices, as well as for forecasting crop and biofuel output and the effects of land conversion on rural infrastructure, including roads and water quality [12]. However, it necessitates comprehensive earth observation data about the historical and existing land-use diversity [13] to enhance the current land management and environmental monitoring systems. For instance, Wulder et al. [14] pointed out that various local, national, and international natural resource management choices require land-use data. Researchers from all over the globe have become interested in machine-learning-based analyses of LULCC and its impacts because of the environmental changes taking place on a global scale [15]. By collecting long-term, high-resolution earth observation (EO) data using orbiting platforms, satellite remote sensing supports effective LULCC monitoring [6] and the analysis of its impact on surface water resources.

Nonetheless, the remote-sensing literature lacks approaches to quantify regional differences in LULCC [16]. Additionally, the ability to measure the spatial and temporal dynamics of LULCC is highly constrained by the lack of high spatial resolution (e.g., IKONOS, GeoEye, and Rapid Eye images), frequently updated datasets for developing countries, and familiarity with robust and up-to-date approaches of LULCC analysis, such as machine learning. For instance, Yang and Huang [6] reported the high scarcity of land-use diversity information produced using machine learning and moderately high-resolution earth observation datasets. In most Sub-Saharan African countries, LULCC and its impacts on surface water bodies have been a serious challenge because it impacts negatively, resulting in severe water scarcity in the area.

For instance, Maviza and Ahmed [17] reported that dense woodland in the Upper Mzingwane sub-catchment of the semiarid region of Matabeleland, South Zimbabwe, decreased by 441 km$^2$ (43.57%), while shrubland, grassland, water bodies, and bare land grew by 237 km$^2$ (10.73%), 185 km$^2$ (4.5%), 14 km$^2$ (26.85%), and 6 km$^2$ (15.09%), respectively. In South Africa, grasslands (27.99%) and shrublands (26.34%) are the two dominant land-use types [18]. In contrast, cropland (15%), barren land (11%), natural forests (16%), and built-up areas (4%) cover comparatively small areas. The total land mass of South Africa is about 1,221,037 (one million two hundred twenty-one thousand and thirty-seven) km$^2$. Most of the LULCC has been triggered due to mining, e.g., in West Africa [19] and Southern African countries. It is also driven by persistent drought, lack of land-use policy, population growth, agricultural land intensification, urbanization, lack of employment, overgrazing, and increasing energy and food demands, among other things [10,20]. Land tenure is also considered an underlying driving force of LULCC [11]. As a result, the potential forest and

water bodies have been shrinking from time to time. This might impact harmonizing the local climate condition [8] and the livelihood system of the inhabitants.

An analysis of LULCC using machine learning is crucial for water resources evaluation, monitoring, and mapping because its impact on the natural environment differs from place to place [19,21]. Spectral differences between different land-use types are used in the classification process. However, the classification and analysis of historical LULCC remained challenging due to the lack of ground-truth data [5] and improved image classification techniques. So far, many studies have focused on LULCC mapping using moderate spatial resolution satellite data (e.g., Landsat 4, 5, 8, and 9; Sentinel; IKONOS; and SPOT). Conversely, a better option for users from developing countries would be to use Landsat images because of their moderate–high spatial resolution, free accessibility, and high temporal resolution. Most existing LULCC studies were analyzed using the supervised maximum likelihood classification techniques in ENVI, ERDAS, ArcGIS, QGIS, and TerrSet, among other geospatial software.

However, the classification and analysis of land-use diversity using a machine-learning approach provides comprehensive information on the various land-use types of the natural environment. For instance, Zurqani et al. [21] applied the RF classifier, one of the most robust machine-learning models, for LULCC analysis in the Savannah River Basin, using Google Earth Engine (GEE). The same authors reported that the major causes of LULCC occurring in the Savannah River Basin were linked to deforestation and reforestation of forest areas during the entire study period. Additionally, four primary limitations need great attention in the existing global land-use maps, e.g., few land classes, absence of temporal updates, coarser resolution, and lower accuracies (~77%) [13]. This topic has received little research due to the challenges in producing high-resolution multitemporal products on a large scale [6]. However, small-to-medium-sized catchments can be thoroughly examined because they have distinct and understandable land-use classes. For comprehending and evaluating LULCC, accurate and up-to-date LULCC information is needed for environmental resources evaluation, monitoring, and modeling. Additionally, it is helpful for various stakeholders to assess future pathways of sustainable land use for food production and nature conservation [22].

This challenge can be resolved using machine-learning models such as RF classifier supported by Google Earth Engine (GEE), ESRI ArcGIS v.10.8.1 software, and the Modified Normalized Difference Water Index (MNDWI). Spectral-index-based surface-water-bodies mapping and analysis methods are robust approaches to discriminate from all non-water bodies' land-use types [23,24]. The spectral water index is a single value that results from arithmetic operations on two or more bands [25]. This helps to carry out a detailed investigation on the surface water content of all water bodies. For instance, the analysis of land-use diversity and its changes strongly impacts regional species richness [2] and water bodies. Therefore, using machine-learning algorithms is crucial for investigating the implications of changing land-use diversity by identifying crucial features from satellite images [26]. Feature selection determines the distinction between out-of-bag (OOB) errors or estimates before and after feature variation. This method helps to focus on essential features that can eliminate the difficulty of interpretation.

Aigbokhan et al. [27] compared the accuracy of four different machine-learning algorithms for land-use diversity mapping, such as RF classifier, Support Vector Machine (SVM), K-Nearest Neighbors (KNNs), and Gaussian Mixture Models (GMMs). The same authors found that RF classified 23% of their study area as bare land, SVM 24%, KNN 24% and GMM 30% for the same land-use type. However, the accuracy of RF was 0.9840, which is higher than that of SVM (0.9780), KNN (0.9641), and GMM (0.9421) machine-learning models. The RF is one of the robust and advanced machine-learning models trained by bootstrapping for land-use diversity mapping using the ground-truth dataset [28]. On GEE, RF java script is enabled to run the model. As a result, the GEE is becoming powerful in identifying and analyzing features at the smallest pixel unit. The traditional per-scene analysis is replaced by per-pixel analysis in GEE because it uses advanced algorithms [13].

Therefore, adequate ground-truth data are needed to ensure the quality of image classification in RF. For instance, Yang and Huang [6] created high-resolution LULC datasets for China based on GEE, and they used a pixel-by-pixel temporal composite to analyze the spectral, phenological, and topographical metrics. Following the covariate predictors, a random subset of the training data is classified using numerous decision trees created by the RF algorithm [29]. RF improves the quality of the image classification process and can quickly manage many features without affecting overall accuracy [5,28]. Moreover, water is an essential natural resource for both livestock and humans. However, it is a serious challenge in the study area due to climate change, less rainfall availability, high population growth, and urbanization, among other factors [30]. Martínez-Núñez et al. [2] analyzed the land-use diversity and Normalized Difference Vegetation Index (NDVI) values for each pixel in GEE. Findell et al. [3] investigated the impact of anthropogenic LULCC on regional climate extremes to characterize the joint temperature–humidity response to land use. Waterlogged areas were successfully extracted using the Modified Normalized Difference Water Index (MNDWI); however, the Normalized Difference Water Index (NDWI) shows the mixing of water with built-up features in the Sri Muktsar Sahib District of Punjab, India [31].

Therefore, there is a need to (i) address research-based essential models for investigating the implication of land-use diversity alteration on surface water resources; and (ii) identify the most accurate surface water detection models. Despite this, the Republic of South Africa's Department of Forestry, Fisheries, and the Environment developed a limited amount of quantitative data on land use at the national level, and reliable machine-learning models such as RF did not support the development of this data. The effects of shifting land-use diversity on surface water resources using remote-sensing models were not examined. Therefore, the novelty of our study is the incorporation of robust approaches for studying the implication of land-use diversity across various district municipalities. The objective of this study was to (i) analyze land-use diversity from the period (1993–2023) by using an RF machine-learning model; (ii) quantify the probability of transfer-out (losses) and transfer-in (gains) rate across each district municipality; (iii) explore the spatiotemporal trends of surface water bodies and determine the implications of changing land-use diversity; and (iv) find the relationship between the Modified Normalized Difference Water Index (MNDWI) and Normalized Difference Water Index (NDWI) for surface water bodies' characterization. This study's findings are crucial to enriching the existing land-use planning and surface water management strategies and reducing the environmental consequence of changing land-use diversity and water supply problems. Additionally, it is imperative to comprehend the spatial distribution of natural resources across each district municipality for planning and conservation. Moreover, it is helpful to realize the performance of GEE-based machine-learning models such as RF for characterizing the implication of changing land-use diversity on surface water bodies.

## 2. Materials and Methods

### 2.1. Study Area

This study was conducted in Gauteng Province, the smallest province in South Africa. It is the economic powerhouse and transportation hub of South Africa [30,32]. It is also one of the mining-dominated provinces of South Africa. The area covers 18,170 km$^2$ (1.22%) and is located between 27°17′15″S to 29°17′25″S and 26°54′45″E to 25°54′40″E (Figure 1). Gauteng Province comprises five districts, namely Tshwane, 6296.2 km$^2$ (34.7%); West Rand, 4084.5 km$^2$ (22.5%); Ekurhuleni, 3458.6 km$^2$ (19%); Sedibeng, 2686.9 km$^2$ (14.8%); and Johannesburg, 1644.0 km$^2$ (9%). There are also eight different metropolitan municipalities in the area. The topography of Gauteng Province ranges from 931 to 1916 m above sea level (m.a.s.l.). The average elevation of Gauteng Province is 1481 m.a.s.l. The mean annual rainfall of the study area is up to 363 mm [30]. The maximum and minimum temperature are 26.3 °C and 20.3 °C, respectively. During the summer, the average temperature can reach 23.3 °C. The study area has two primary climatic seasons. These are

the dry season (April–October) and winter (May–August). Based on the Soil and Terrain Database of Southern Africa (SOTERSAF version 1.0), nine dominant soil types exist in the study area. Lixisols are the most dominant soil type, covering 6895.3 km$^2$ (37.9%), while Solonetz covers 68 km$^2$ (0.4%). The remaining soil types include Acrisols, covering 3656.1 km$^2$ (20.1%); Leptosols, 2924.2 km$^2$ (16.1); Plinthosols, 2458.3 km$^2$ (13.5%); Vertisols, 1150.2 km$^2$ (6.3%); Luvisols, 448.3 km$^2$ (2.5%); Nitisols, 441.3 km$^2$ (2.4%); and Regosols, 100 km$^2$ (0.6%). The remaining part of the area is covered by a water body, 28.4 km$^2$ (0.2%). About 15,488,137 people live in Gauteng Province and are receiving water from the Vaal Dam [33,34]. However, the area is highly characterized by crystalline nature rocks, making it challenging to extract adequate groundwater [35]. Additionally, it is impacted by a variety of hydrogeological factors, including drainage density, lithology, slope, and geomorphology [36].

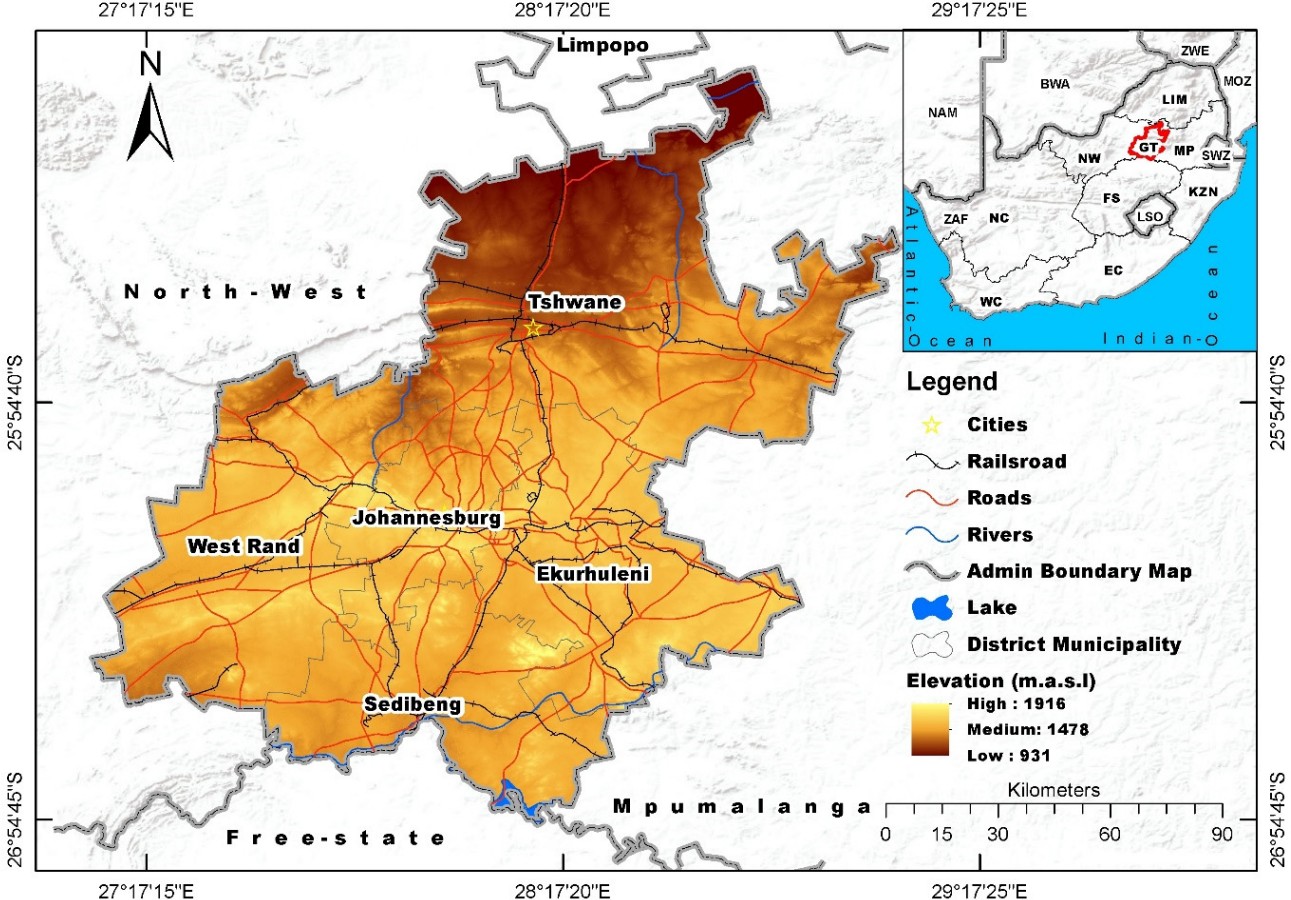

**Figure 1.** Location of the study area.

### 2.2. Image Acquisition

One of the most frequently utilized data sources for investigating the implications of changing land-use diversity on surface water resources is Landsat imagery. The Landsat multispectral and multitemporal imageries collection 2, level 2, acquired during daytime, used at moderate spatial resolution for 1993, 2003, 2013, and 2022 (Table 1) downloaded from four different scenes (tiles), namely path 170/row 77, 170/78, 170/79, and 171/78. The availability of those medium-spatial-resolution satellite imageries drew the attention of scholars to investigate the diversity and changes in land use in the landscape. Several scholars have also successfully used the datasets to evaluate LULCC and other related activities [18].

**Table 1.** Kappa coefficient values threshold.

| Level of Agreement | Kappa Coefficient |
|---|---|
| Excellent | 0.8–1.0 |
| Good | 0.6–0.8 |
| Moderate | 0.4–0.6 |
| Weak | 0.2–0.4 |
| Bad | −1.0–0.2 |

Training samples were also collected by supporting false-color-composite and visual-image-interpretation techniques from all study areas' land-use types in the GEE platform. GEE has the benefit of simultaneously processing large pools of satellite imagery quickly, and it has powerful computational algorithms [13]. According to Zhang et al. [37], there are two required methods for gathering ground-truth data in a remote-sensing environment: (i) using a false-color composite supported by the visual-image-interpretation approach and (ii) creating sample points automatically from satellite images that are prepared for classification. Therefore, manual interpretation and/or semi-automated classification backed by field surveys are/is required for the identification, monitoring, mapping, and analysis of land-use diversity [26]. Our research used the false-color-composite method supported by the visual-image-interpretation method. After that, we gathered samples from each land-use type separated by their spectral values. Studies also indicated that visual-image-interpretation-based approaches to ground-truth data collection are higher in quality than sample points that are randomly created from satellite images. The consistency of postprocessing land-use datasets depends on these training samples [6].

### 2.3. Image Preprocessing, Interpretation, and Analysis

The primary prerequisites for remote-sensing products; processing and the LULCC research are (i) image preprocessing and enhancement, (ii) appropriate image-classification-technique selection, and (iii) collection of reference datasets for accuracy and validation. The geometric, radiometric, and atmospheric errors had to be fixed to enhance the visibility of each pixel and eliminate positional inclination (if any) before any research was performed. As part of this requirement, the issue of cloud cover (shadow) and its effect is not new [38], but it is critical in satellite-image analysis based on their spectral signatures. For instance, cloud cover can significantly hinder the analysis of crop or vegetation growth [39]. Therefore, the effect of cloud, haze, shadow, or other disturbances in the input images must be corrected to optimize classification accuracy [13]. In our study, we used high qualities of Landsat imageries Level 2, which are atmospherically and radiometrically corrected and enhanced images. Furthermore, geometric corrections should be applied to remove distortions caused by the sensor or the earth's rotation. The Universal Transverse Mercator (UTM) Zone 35 South projection and World Geodetic System 1984 (WGS84) data were applied for all inputs that we used. Studies have shown that RF has successfully processed and analyzed remote-sensing products even with more pronounced noise effects. Zurqani et al. [21] reported that LULCC was evaluated across the study area, using pre-processed Landsat imagery made accessible by GEE. A quick analysis utility for LULCC is provided by GEE and was analyzed based on Table 2. Following that, a pixel-by-pixel study was performed using the spectral values of each individual pixel (Equation (1)):

$$g_i(x) = 1np(w_i) - \frac{1}{2}1n\left|\sum_i\right| - \frac{1}{2}(x - m_i)^T \sum_i{}^{-1}(x - m_i) \qquad (1)$$

where $i$ = the $i^{\text{th}}$ class (e.g., class 1, 2, 3…. $n$), $x$ = $n$–dimensional data (where $n$ is the number of bands), $p(w_i)$ = probability that a class occurs in the image and is assumed the same for all classes, $\left|\sum_i\right|$ = determinant of the covariance matrix of the data in a class, $\sum_i{}^{-1}$ = the inverse of the covariance matrix of a class, and $m_i$ = mean vector of a class.

**Table 2.** Spectral index for surface water body detection and analysis.

| Surface Water Detection Indices | Landsat 4–5 (TM) and 8 OLI | Reference |
| --- | --- | --- |
| NDWI | Green–NIR/Green + NIR | McFeeters (1996) [40] |
| MNDWI | Green–SWIR1/Green + SWIR1 | Xu (2005, 2006) [24,41] |

The spectral values of each pixel were analyzed using the N × 300 ground-truth data to improve the classification processes and accuracy of the findings. In this case, N refers to the total number of land-use types in the study area. Gbedzi et al. [19] reported that about 50 ground-truth datasets per land-use type were used to classify the satellite imageries and assess the level of accuracy. There are 500 trees in the RF classification for categorizing seven land-use types [29]. Yang and Huang [6] reported that the accuracy level of China land-cover datasets is evaluated by applying confusion matrixes, namely producer's accuracy, consumer's accuracy, overall accuracy, and F-score. F-score is one of the machine-learning measures that help assess the RF model's precision. The proportion of all instances that are classified by the model is known as classification accuracy [42]. The minimum level of the overall accuracy should be at least 85 percent [43]. In this study, we analyzed the confusion matrixes, namely consumer's accuracy (CA), producer's accuracy (PA), overall accuracy (OA), F-score based on the Yang and Huang [6] formula, and Kappa coefficient (Table 1), as follows (Equations (2)–(6)):

$$CA = x_{ii}/x_{i+} \times 100 \tag{2}$$

$$PA = x_{ii}/x_{+i} \times 100 \tag{3}$$

$$OA = D/V \times 100 \tag{4}$$

$$\text{F} - \text{score} = 2 \times (PA \times CA)/(PA + CA) \tag{5}$$

$$\hat{K} = \frac{N\sum_{i=1}^{k} x_{ii} - \sum_{i=1}^{k}(x_{i+} \times x_{+i})}{N^2 - \sum_{i=1}^{k}(x_{i+} \times x_{+i})} \tag{6}$$

where $\hat{K}$ is the Kappa coefficient, or K-coefficient; $x_{ii}$ is the total number of observations in row *i* and column *i*; $x_{i+}$ and $x_{+i}$ are the marginal totals of row *i* and column *i*, respectively; *N* is the total number of observations; *D* is the total number of correct pixels (diagonal); *V* is the total number of pixels in the error matrix; F-score is the harmonic mean of PA and CA; PA is the producer's accuracy; and CA is the consumer's accuracy.

To realize the spatial and temporal patterns of LULCC in the study area, a post-classification change detection analysis was performed pixel-by-pixel between the final and initial study years. This analysis was carried out by deducting the spatial area coverage of the final year from the initial year. This change-detection method was proven to be the most effective technique because data from two periods are separately classified, thereby minimizing the problem of normalizing for atmospheric and sensor differences [44].

*2.4. Surface Water Content Detection, Analysis, and Mapping*

The surface water bodies of Gauteng Province were identified to investigate their history and existing status, using Landsat 4–5 TM and 8 OLI satellite images. We also evaluated the water content of the study area for proper planning and management. The Modified Normalized Difference Water Index (MNDWI) and Normalized Difference Water Index (NDWI) were used based on Table 2. Both indices are novel for surface water body detection and water content analysis. They are also useful methods for determining water quality since they exclude soil and terrestrial vegetation features within a water body [40].

The MNDWI overcomes the limits of the Normalized Difference Water Index (NDWI) regarding soil and built-up areas but still has proven to be effective in separating water bodies and vegetation [45]. The MNDWI improves the detection of water-content information and eliminates shadow noise in areas where built-up lands are a dominant cover type than the Normalized Difference Water Index (NDWI) [41]. The MNDWI also provides more detailed and higher-quality water-content information than the NDWI [46,47]. In this study, we employed both indices to verify the water content of the study area (Table 2; Figure 2).

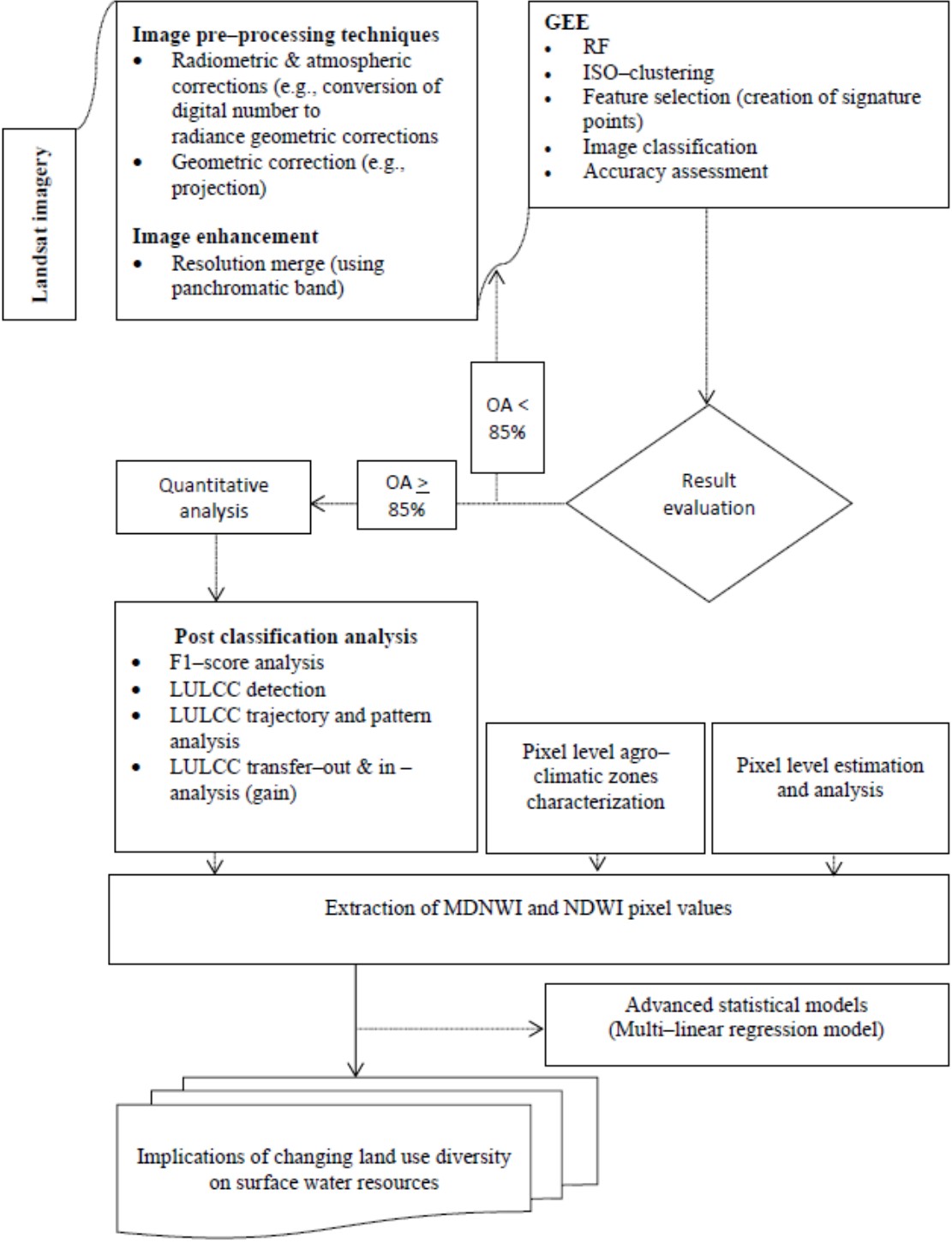

**Figure 2.** Impacts of changing land-use variety on surface water bodies based on GEE.

For the NDWI estimation, the Green and NIR represent bands 2 and 4 in Landsat 4–5 TM; however, they signify bands 3 and 5 in Landsat 8 OLI products. Furthermore, for the analysis of the MNDWI, the Green and SWIR1 represent bands 2 and 5 in Landsat 4–5 TM and bands 3 and 6 in Landsat 8 OLI. The MNDWI uses green and shortwave infrared (SWIR–1) bands to point out water bodies, while the NDWI uses green and near-infrared (NIR) bands. The value of the MNDWI and NDWI ranges from −1 to +1. Built-up areas, soil, and vegetation have negative values due to their higher reflectance, while surface water bodies have positive values because of lower reflectance in the SWIR band [31].

*2.5. The Statistical Relationship between the MNDWI and NDWI*

We applied the Pearson correlation coefficient (r) to investigate the relationships between the NDWI and MNDWI (Equation (7)). In most cases, correlations are utilized to analyze bivariate relationships between measured variables. In this study, we generated the values of each pixel found within a 5, 10, and 15 km radius distance to compare the results from the center of the study site to assess the relationships of each factor. This increases the processing capability of our computer and Minitab version 16 statistical software. The Pearson correlation coefficient values range from −1 to +1. Bolboaca and Jäntschi [48] reported that if the variables have a value of +1, an increasing relationship completely associates them; if they have a value of −1, they are perfectly related by a decreasing relationship, and if they have a value of 0, they are not linearly related to each other. If the correlation coefficient is more than 0.8, the correlation is strong; if it is lower than 0.5, the correlation is weak. In several studies, the Pearson correlation matrix was used to assess the relationships between the NDWI and MNDWI [2,49].

$$r = \frac{n(\sum xy) - (\sum x)(\sum y)}{\sqrt{\left[ n\sum x^2 - \left( \sum x \right)^2 \right] \left[ n\sum y^2 - \left( \sum y \right)^2 \right]}} \tag{7}$$

where $r$ = Pearson correlation coefficient, $n$ = no. of pairs, $\sum xy$ = sum of $x$ and $y$, $(\sum x) = sum\ of\ x\ values, (\sum y) = sum\ of\ y\ values$, $\sum x^2$ = sum of the squared $x$ value, and $\sum y^2$ = sum of the squared $y$ value.

**3. Results and Discussion**

*3.1. Land-Use Diversity Analysis at the Province Level*

Using the RF classifier machine-learning model, we found nine different land-use diversity classes in the study area, namely WB, MQ, DF, GL, ShL, BuA, WeL, AL, and BL (Figure 3a–d), with high F-score values ranging from 72.3 to 100% during the period from 1993 to 2022 (Table 3). The results of CA and PA also show a strong agreement with the F-score values of each land-use type (Table 3). These values were validated using the OA and K-coefficient. The OA and K-coefficient values of the study area for the years 1993, 2003, 2013, and 2022 were 92, 93, 95, and 90%, respectively, with greater K-coefficient values of 90, 91, 93, and 87% (Table 3). Mawasha and Britz [50] reported an OA of 85.9, 87.5, and 92.5% for the years 1987, 2001, and 2015, respectively, with a Kappa coefficient of 81.3, 83.3, and 90% in the Jukskei River catchment, Gauteng, South Africa. Azzari and Lobell [13] stated an OA of 89% for cropland and non-cropland in Zambia. According to Zurqani et al. [21], the OAs for the years 1999, 2005, 2009, and 2015 were 79.18, 79.41, 76.04, and 76.11%. However, Anderson [43] underlined that an acceptable OA value should be at least 85%. Our findings were above the minimum standard and acceptable range in this case. We also proved this value with the F-score value of each land-use type in our study area. Various studies also indicated that an F-score value above the average (50%) indicates perfect precision and is considered to be in higher agreement (Table 3). Table 3 shows our study area's detailed CA, PA, F-score, OA, and K-coefficient analysis from 1993 to 2022.

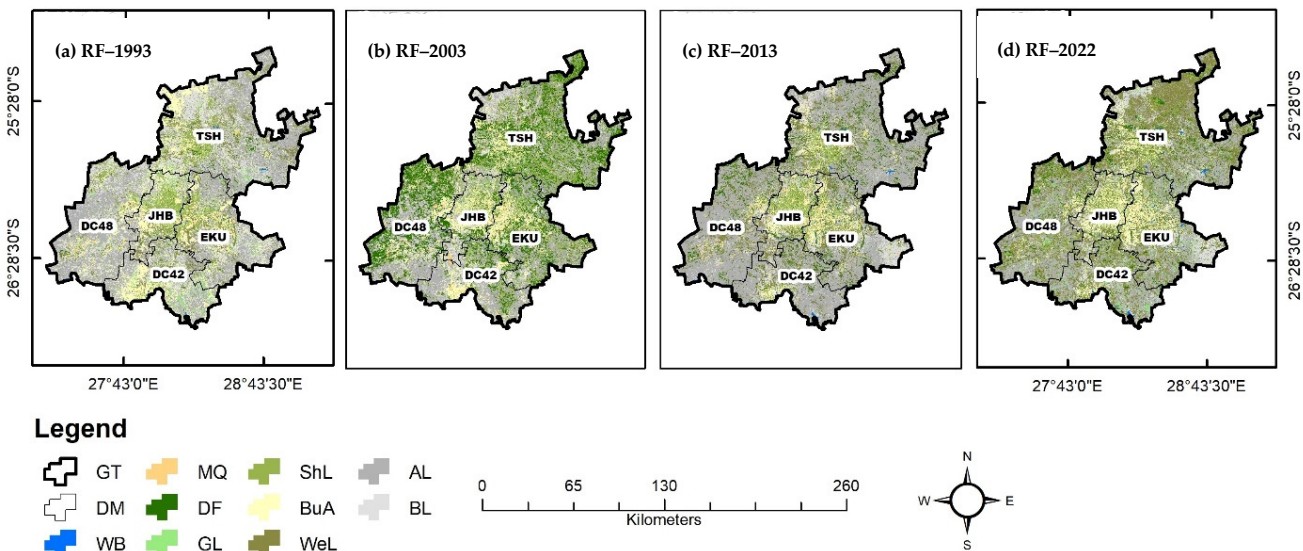

**Figure 3.** (**a**–**d**) Spatial–temporal trends of land-use diversity in Gauteng Province, 1993–2022. **GP**, Gauteng Province; **DM**, district municipality; **WB**, water body; **MQ**, marine quarry; **DF**, dense forest; **GL**, grassland; **ShL**, shrub/bushland; **BuA**, built-up area; **WeL**, wetland, including rivers; **AL**, arable land; **BL**, barren land (open land); **TSH**, Tshwane; **JHB**, Johannesburg Metropolitan City; **DC48**, West Rand; **DC42**, Sedibeng; **EKU**, Ekurhuleni.

**Table 3.** The detailed analysis of CA, PA, F-score, OA, and K-coefficient value from 1993 to 2022.

| LC | 1993 | | | 2003 | | | 2013 | | | 2022 | | |
|---|---|---|---|---|---|---|---|---|---|---|---|---|
| | **CA%** | **PA%** | **F-Score** | **CA%** | **PA%** | **F-Score** | **CA%** | **PA%** | **F-Score** | **CA%** | **PA%** | **F-Score** |
| WB | 93.8 | 96.8 | 95.2 | 97.1 | 100.0 | 98.5 | 100.0 | 100.0 | 100.0 | 91.2 | 100.0 | 95.4 |
| MQ | 86.8 | 80.5 | 83.5 | 100.0 | 100.0 | 100 | 97.8 | 95.8 | 96.8 | 84.6 | 83.0 | 83.8 |
| DF | 98.5 | 100.0 | 99.3 | 87.8 | 98.5 | 92.8 | 97.2 | 99.7 | 98.5 | 90.7 | 98.2 | 94.3 |
| GL | 88.9 | 88.9 | 88.9 | 97.4 | 100.0 | 98.7 | 97.8 | 93.8 | 95.7 | 93.6 | 100.0 | 96.7 |
| ShL | 90.7 | 97.5 | 94.0 | 95.8 | 65.7 | 78.0 | 97.0 | 86.5 | 91.4 | 86.8 | 82.5 | 84.6 |
| BuA | 89.1 | 87.5 | 88.3 | 100.0 | 100.0 | 100.0 | 89.2 | 94.3 | 91.7 | 75.0 | 69.8 | 72.3 |
| WeL | 94.9 | 92.5 | 93.7 | 94.6 | 70.0 | 80.5 | 89.1 | 83.7 | 86.3 | 85.0 | 58.6 | 69.4 |
| AL | 71.4 | 80 | 75.5 | 90.2 | 84.6 | 87.3 | 92.1 | 90.6 | 91.3 | 93.6 | 84.6 | 88.9 |
| BL | 90.2 | 74 | 81.3 | 92.4 | 93.8 | 93.1 | 81.3 | 81.3 | 81.3 | 95.4 | 95.4 | 95.4 |
| OA% | 92 | | – | 93 | | | 95 | | – | 90 | | – |
| K% | 90 | | | 91 | | | 93 | | | 87 | | |

In GP, AL was the dominant land-use type in 1993. This land-use type covered an area of 6156.0 km$^2$ (33.88%) in 1993 (Figure 3a). Following by AL, BL covered an area of 4565.60 (25.13%); BuA, 2713.40 (14.93%); ShL, 2071.20 (11.40%); WeL, 1646.90 (9.06%); GL, 553.30 (3.05%); MQ, 213.80 (1.18%); DF, 182.60 (1%); and WB, 67.10 km$^2$ (0.37%) (Table 4). However, AL, BL, ShL, and BuA declined annually by 365.4, 293.2, 181, and 165.4 km$^2$, respectively, during the period of 1993–2003 (Figure 4a). In the same period, an increased trend was observed in DF, WB, GL, WeL, and MQ by 2.8, 37.1, 297.2, 326.7, and 341.3 km$^2$, respectively. In the year 2003, WeL was the predominant land use, covering around 4913.5 km$^2$ (27%) of the total area (Table 4; Figure 3b). The remaining land-use types, namely MQ, GL, AL, BL, BuA, WB, ShL, and DF covered an area of 3627.1 (20%), 3525.2 (19.4%), 2502.5 (13.8%), 1633.10 (9%), 1059.6 (5.8%), 437.6 (2.4%), 260.8 (1.4%), and 210.7 km$^2$ (1.2%), respectively (Table 4). The 2013 land-use diversity of the study area shows that AL was still covering the highest occupancy rate, i.e., 7569.6 km$^2$ (41.7%), in comparison to other land-use categories in the area (Figure 3c). This land-use type shows an increased trend from its earlier coverage by 5067.1 km$^2$ (Table 4). The remaining land-use types included WeL, which covered 3878.8 (21.3%); BL, 1876.4 (10.3%); ShL, 1433.1 (7.9%); DF, 1426.5 (7.9%); BuA, 1053.9 (5.8%); MQ,

593.8 (3.3%); GL, 250.7 (1.4%); and WB, 87.1 km$^2$ (0.5%) (Table 4; Figure 3c). Similarly, in the period 2003–2013, GL, MQ, WeL, WB, and BuA showed an annual decline of 327.5, 303.3, 103.5, 35.1, and 0.6 km$^2$, respectively (Figure 4b). Moreover, the 2022 land-use diversity of the study area (Figure 3d) indicated that AL occupied the largest area, i.e., 6241.5 (34%). However, WeL covered 4243 (23%); ShL, 2395.8 (13%); BL 1654.7 (9.1%); BuA, 1490.7 (8.2%); DF, 1041.4 (5.7%); GL, 715.8 (3.9%); MQ, 281.6 (1.5%); and WB, 105.7 km$^2$ (0.6%) (Table 4). In the years 2013–2022, AL, DF, MQ, and BL also showed an annual decline trend of 147.6, 42.8, 34.7, and 24.6 km$^2$, respectively (Figure 4c). However, the extent of WB, WeL, BuA, GL, and ShL intensified annually by 2.1, 40.5, 48.5, 51.7, and 107 km$^2$, respectively. This study reported that during the last three decades (1993–2022), only BL and BuA declined by 100.4 and 42.2 km$^2$, respectively (Figure 4d). The presence of high vegetation in the area was strongly affecting the classification of BuA. The BuA was heavily dominated by vegetation in the majority of GP district municipalities, including the metropolitan center of Johannesburg. As a result, the spatial extent of BuA is less than the previous years. The remaining land-use types, namely WB, MQ, AL, GL, ShL, DF, and WeL, have expanded annually by 1.3, 2.3, 2.9, 5.6, 11.2, 29.6, and 89.5 km$^2$, respectively. Those land uses may affect the security of surface water bodies in the area. Abiye [35] stated that the economic sustainability of the area is hampered by water insecurity (e.g., surface water), which is brought on by factors such as population growth, expansion of industrial, agricultural, and mining sectors, among others. In addition to this, the hydrological cycle may also be significantly impacted by changes in water quality and surface runoff [33].

**Table 4.** Land-use diversity classes in the study area from 1993 to 2022.

| L-Class | 1993 | | 2003 | | 2013 | | 2022 | |
|---|---|---|---|---|---|---|---|---|
| | Area in km$^2$ | % | Area in km$^2$ | % | Area in km$^2$ | % | Area in km$^2$ | % |
| MQ | 213.80 | 1.18 | 3627.10 | 20.00 | 593.80 | 3.30 | 281.60 | 1.50 |
| DF | 182.60 | 1.00 | 210.70 | 1.20 | 1426.50 | 7.90 | 1041.40 | 5.70 |
| GL | 553.30 | 3.05 | 3525.20 | 19.40 | 250.70 | 1.40 | 715.80 | 3.90 |
| ShL | 2071.20 | 11.40 | 260.80 | 1.40 | 1433.10 | 7.90 | 2395.80 | 13.00 |
| BuA | 2713.40 | 14.93 | 1059.60 | 5.80 | 1053.90 | 5.80 | 1490.70 | 8.20 |
| WeL | 1646.90 | 9.06 | 4913.50 | 27.00 | 3878.80 | 21.30 | 4243.00 | 23.00 |
| AL | 6156.00 | 33.88 | 2502.50 | 13.80 | 7569.60 | 41.70 | 6241.50 | 34.00 |
| BL | 4565.60 | 25.13 | 1633.10 | 9.00 | 1876.40 | 10.30 | 1654.70 | 9.10 |
| Surface water bodies' coverage | | | | | | | | |
| WB | 67.10 | 0.37 | 437.60 | 2.40 | 87.10 | 0.50 | 105.70 | 0.60 |
| Total area coverage | 18,170.00 | 100.00 | 18,170.0 | 100.00 | 18,170.0 | 100.00 | 18,170.0 | 100.00 |

Mawasha and Britz [50] reported an increase in BuA and BL by 427.13 (56.2%) and 62.251 km$^2$ (8.2%), whereas a decrease of vegetation cover by 74.55 (9.8%) and 197 km$^2$ (25.8%), respectively, was observed for the period of 1987–2015. The decline in AL impacts the accessibility of food for both humans and livestock. This has been proven by [51] because livestock occupies 70% of all AL and GL. The decreased trend of WB may harm AL productivity [52]. Additionally, the reduction of MQ could influence the economic growth of GP. However, unwisely treated MQ causes negative environmental effects linked with the contamination of surface and subsurface water resources [53]. Therefore, it is important to undertake proper MQ waste management to safeguard the local water bodies and other natural resources.

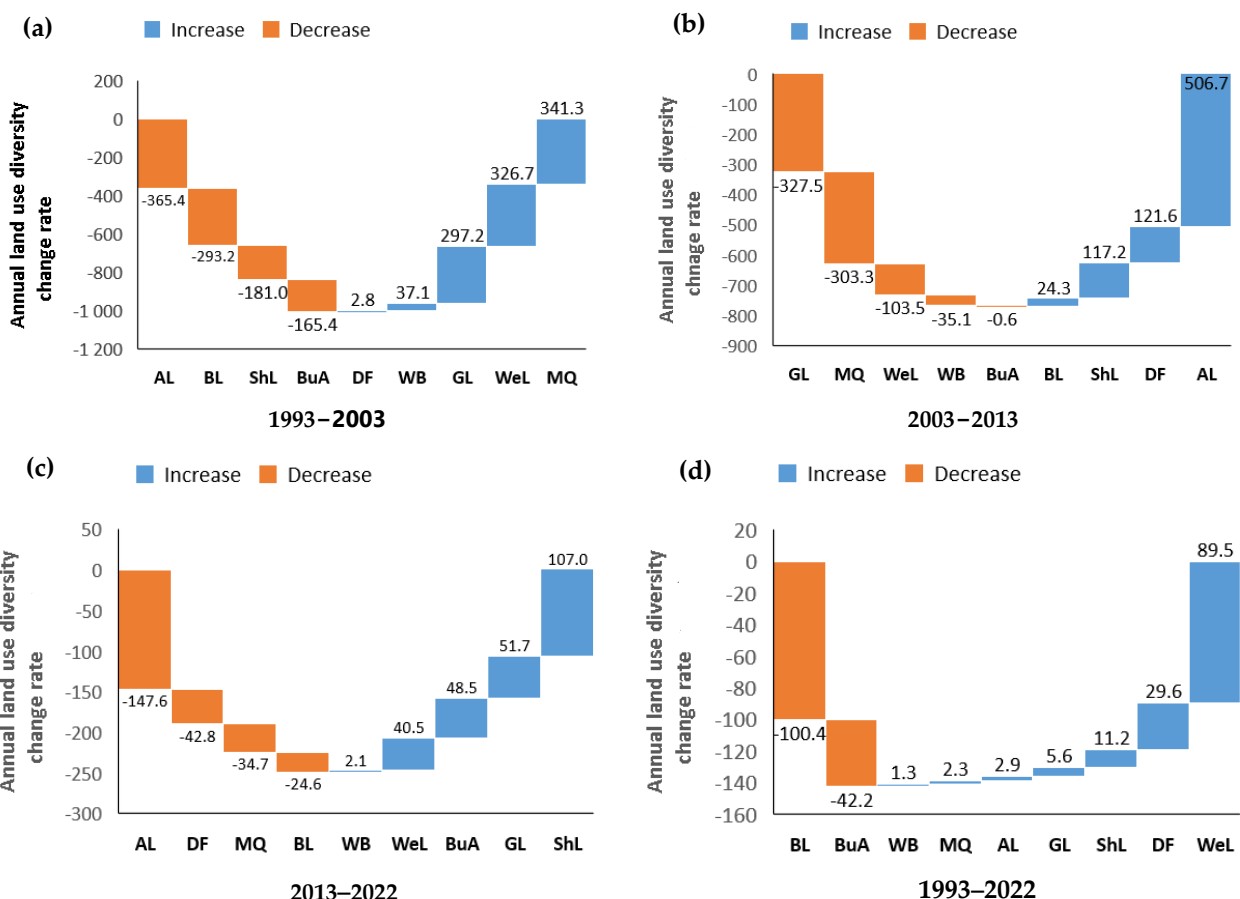

**Figure 4.** (**a–d**) Annual rate of land-use diversity change: transfer-out (loss) and transfer-in (gain) analysis in km$^2$ for GP from 1993 to 2022.

### 3.2. Land-Use Diversity Class Analysis at the District Municipality Level

The smallest portion of the Ekurhuleni (EKU) District (19%) of Gauteng Province was covered by WB (0.5), DF (0.9), and MQ (2.1%) in the year 1993 (Figure 3a). However, the most dominant land-use types of the area were AL, which covers 1068.4 (30.9%); BL, 770.5 (22.3%); BuA, 572.1 (16.5%); ShL, 360.4 (10.4%); and GL, 107 km$^2$ (3.1%). In 2003, the spatial coverage of all land-use types changed to different spatial extents. For instance, WeL covered an area of 953.8 (27.6%); GL, 703.9 (20.4%); MQ, 683.2 (19.8%); BL, 398.3 (11.5%); AL, 306.8 (8.9%); BuA, 208.1 (6%); WB, 108.8 (3.1%); DF, 45.3 (1.3%); and ShL, 50.3 km$^2$ (1.5%) (Figure 3b). In 2013, a remarkable shift in various land uses was observed. During this time, the largest portion of the study area was covered by AL, 1231.6 km$^2$ (35.6%). Following AL, WeL covered 612.8 (17.7%), BL covered 623.7 (18%), BuA covered 313.1 (9.1%), ShL covered 256.9 (7.4%), DF covered 190.1 (5.5%), MQ covered 151.8 (4.4%), GL covered 54.9 (1.6%), and WB covered 23.7 km$^2$ (0.7%) (Figure 3c). Similarly, in the year 2022, AL covered 927.8 (26.8%), BL covered 560.3 (16.2%), BuA covered 504.9 (14.6%), WeL covered 743.4 (21.5%), ShL covered 327.3 (9.5%), GL covered 163.2 (4.7%), DF covered 127.7 (3.7), MQ covered 74.5 (2.2%), and WB covered 29.8 km$^2$ (0.9%) (Figure 3d). Moreover, EKU experienced a significant loss (decrease) and gains (increase) in potential land use in 1993, 2003, 2013, and 2022. Supplementary Figure S1a indicates that AL, BuA, BL, and ShL annually declined by 76.2, 36.4, 37.2, and 31 km$^2$, respectively, from 1993 to 2003. In the same period, the land-use types MQ, GL, WeL, WB, and DF increased by 61, 59.7, 49.4, 9.2, and 1.4 km$^2$. In 2003–2013, the previously increased land-use types declined in their spatial coverage. For instance, GL, MQ, WeL, and WB were annually diminishing by 64.9, 53.1, 34.1, and 8.5 km$^2$, respectively (Supplementary Figure S1b). On the contrary, AL, BL, ShL, DF, and BuA intensified yearly by 92.5, 22.5, 14.5, and 0.5 km$^2$, respectively. Similarly, in 2013–2022,

WB showed a slight annual increase (gains) of 0.6 km$^2$. During this time, the corresponding annual increases for BuA, WeL, GL, and ShL were 19.2, 13, 10.8, and 7.1 km$^2$, respectively (Supplementary Figure S1c). However, there was a noticeable decrease in AL, MQ, BL, and DF of 30.4, 7.7, 6.3, and 6.2 km$^2$, respectively. In general, an increased trend of MQ, WB, GL, DF, and WeL by 0.1, 1.5, 6.2, 10.8, and 31.5 km$^2$, respectively, was observed in EKU from the year 1993 to 2022. In the same period, the spatial extent of BL, AL, BuA, and ShL declined annually by 23.4, 15.6, 7.5, and 3.7 km$^2$, respectively (Supplementary Figure S1d).

The Johannesburg (JHB) Metropolitan City of Gauteng Province, which occupied only 9%, was dominated by ShL during 1993 (Figure 3a). The largest portion of the city was covered by healthy vegetation, including BuA, throughout the study years (Figure 3a). During this time, ShL covered an area of 385.6 km$^2$ (23.5.2%), while the land-use type BuA covered 420.9 (25.6%), BL covered 259.1 (15.8%), AL covered 214.1 (13%), WeL covered 190 (11.6%), GL covered 104.3 (6.3%), MQ covered 48.5 (2.9%), WB covered 2.8 (0.2%), and DF covered 18.7 km$^2$ (1.1%). Additionally, in 2003, BL covered the largest portion of the city. About 350.6 km$^2$ (21.3%) was covered by BL, WeL covered 272 (16.5%), AL covered 257 (15.7%), GL covered 248 (15.1%), MQ covered 164.3 (10%), WB covered 108.3 (6.6%), ShL covered 77 (4.7%), BuA covered 83.1 (5.1%), and DF covered 78.2 (4.8%) (Figure 3b). This coverage was reverted in 2013 because of the land-use dynamics. During this time, the largest portion of JHB city was occupied by ShL 352.8 km$^2$ (21.5%). Following this, BuA was the second dominant land-use type and covered an area of 279.1 (17%), WeL covered 249.8 (15.2%), AL covered 242.3 (14.7%), BL covered 260.3 (15.8%), MQ covered 128.5 (7.8%), DF covered 70.9 (4.3%), GL covered 54.9 (3.3%), and WB covered 5.3 km$^2$ (0.3%) (Figure 3c). However, by 2022, BuA had replaced ShL as the area's primary land-use type, covering 450 km$^2$ (27.4%). ShL covered 219.9 (13.4%), BL covered 210.1 (12.8%), AL covered 195 (11.9%), WeL covered 345 (21%), GL covered 85.3 (5.2%), MQ covered 69.5 (4.2%), DF covered 63.3 (44.6%), and WB covered 5.9 km$^2$ (0.4%) (Figure 3d). Moreover, in the period 1993–2003, most of the land-use types, namely AL, DF, WeL, BL, WB, MQ, and GL, showed an annual increase (gains) in spatial coverage by 4.3, 6, 8.2, 9.2, 10.6, 11.6, and 14.4 km$^2$. However, both ShL and BuA showed a decline of 33.9 and 33.8 km$^2$. This study strongly argues that the effect of intense vegetation covering most BuA shows a declining trend because most BuA pixels were dominantly covered by ShL and DF (Supplementary Figure S2a). In 2003–2013, significant land-use types declined in their spatial extent (Supplementary Figure S2b). For instance, GL decreased by 19.4, WB by 10.3, BL by 9, MQ by 3.6, WeL by 2.2, AL by 1.5, and DF by 0.7 km$^2$ per year (Supplementary Figure S2b). However, upon the loss of other land-use types, ShL and BuA exhibited increases of 27.6 and 19.6 km$^2$, respectively. Except for BuA, WeL, GL, and WB, all other land-use classifications decreased by various degrees between 2013 and 2022 (Supplementary Figure S2c). The size of the other land-use types, namely BuA, WeL, GL, and WB, increased by 17.1, 9.5, 9, and 0.1 km$^2$ annually. Conversely, ShL, MQ, BL, AL, and DF shrunk by 13.3, 5.9, 5 km$^2$, 4.7, and 0.8 km$^2$ per annum, respectively. Furthermore, the coverage of ShL, which is crucial for lessening the effects of climate change at the local level, generally decreased during the past three decades (1993–2022) in the metropolitan city of JHB (Supplementary Figure S2d). During this period, ShL declined annually by 18.4, BL by 5.4, AL by 2.1, and GL by 2.1 km$^2$. However, the spatial extent of WeL, DF, BuA, MQ, and WB increased by 17.2, 5.0, 3.2, 2.3, and 0.3 km$^2$ annually.

The Tshwane (TSH) District of Gauteng Province, which covers 34.7%, was dominated by AL during 1993 (Figure 3a). This land-use type occupied 2143.6 km$^2$ (34%) of the district. Following this, BL covered 1821.9 (28.9%), ShL covered 809.5 (12.9%), BuA covered 664.1 (10.5%), WeL covered 608.7 (9.7%), GL covered 105.9 (1.7%), MQ covered 24.5 (0.4%), and WB covered 15.9 km$^2$ (0.3%). WB occupied the smallest portion of TSH. However, most land-use types were changing into other land-use types in 2003. For instance, GL replaced the dominance of AL in 2003. This might have occurred because of the similarity of early stage crop growth with GL and GL's expansion within AL. Following this, WeL covered an area of 1522.2 (24.2%); MQ, 1395.2 (22.2%); AL, 781.1 (12.4%); BuA, 354.8 (5.6%); BL, 306 (4.9%); WB, 81.8 (1.3%); DF, 53 (0.8%); and ShL, 53.9 km$^2$ (0.9%) (Figure 3b). In 2013,

AL dominated the entire land mass of TSH (Figure 3c). The spatial extent of GL dropped to 53.4 km$^2$ (0.8%) due to the expansion of other land-use types, such as AL. During this time, AL covered 3249.3 (51.6%), WeL covered 1181 (18.8%), BL covered 669.4 (10.6%), DF covered 456.2 (7.2%), ShL covered 355.3 (5.6%), BuA covered 201.4 (3.2%), MQ covered 106 (1.7%), GL covered 53.4 (0.8%), and WB covered 24 km$^2$ (0.4%). In 2022, WeL overtook the dominance of AL by covering an area of 2310.9 (36.7%); AL, 1656.8 (26.3%); ShL, 817.6 (8.5%); DF, 389.2 (6.2%); BuA, 349.1 (5.5%); GL, 151.8 (2.4%); MQ, 47 (0.7%); and WB, 36.7 km$^2$ (0.6%) (Figure 3d). In addition, during 1993–2003, the extent of BL declined annually by 151.6 km$^2$ (Supplementary Figure S3a). The decrease in BL in the TSH District may provide opportunities to expand other crucial land-use types, such as AL, ShL, BuA and DF. AL diminished annually by 136.3 km$^2$. The decline in AL may also critically affect the agricultural production and productivity of the area. Moreover, ShL, BuA, and DF were reduced annually by 75.6 km$^2$, 30.9 km$^2$, and 4.9 km$^2$, respectively. The decline in ShL and DF has negatively impacted the area's climate condition. Both ShL and DF are vital for climate regulation. During the same period, TSH benefited from the increase in GL, MQ, WeL, and WB by 164.2, 137.1, 91.4, and 6.6 km$^2$, respectively. In the period 2003–2013, the increases, as well as decreases, of earlier land-use changed on various scales (Supplementary Figure S3b). For instance, the spatial extent of GL, MQ, WeL, BuA, and WB declined by 169.5, 128.9, 34.1, 15.3, and 5.8 km$^2$, respectively. On the other hand, AL, DF, BL, and ShL gained 246.8, 40.3, 36.3, and 30.1 km$^2$ annually. In the years 2013–2022, AL, BL, DF, and MQ showed a tendency to decline by 159.3, 13.3, 6.7, and 5.9 km$^2$, respectively (Supplementary Figure S3c). Nonetheless, WeL, ShL, BuA, GL, and WB showed a positive trend in terms of their spatial coverages and grew by 113, 46.2, 14.8, 9.8, 1.3, and 1.3 km$^2$, respectively. Overall, we observed that BL, AL, and BuA declined by 142.8, 54.1, and 35 km$^2$ from the year 1993 to 2022 (Supplementary Figure S3d). The decline in BuA during the study years is linked to the predominance of ShL and DF. Nevertheless, WeL, DF, GL, MQ, WB, and ShL improved annually by 189.1, 31.9, 5.1, 2.5, 2.3, and 0.9 km$^2$, respectively.

The West Rand (DC 48) District of Gauteng Province, which covers about 22.5%, was dominated by AL during 1993 (Figure 3a). AL occupied 1956.3 km$^2$ (47.9%) of the entire land mass within this period. Following this, BL was the second largest DC48 coverage. It was sized 1096.4 km$^2$ (26.8%), whereas BuA covered 417.9 (10.2%), ShL covered 274.6 (6.7%), WeL covered 174.8 (4.3%), GL covered 88.7 (2.2%), MQ covered 53.5 (1.3%), DF covered 18.3 (0.4%), and WB covered 4 km$^2$ (0.1%). In the year 2003, a significant change in almost all land-use types occurred. For instance, MQ expanded by 1033 km2 from its earlier coverage, while WeL intensified by 908.7, GL by 383.8, and WB by 67.5 km$^2$. However, the extent of AL diminished by 1206.2, BL by 819.6, ShL by 250.9, BuA by 109.6, and DF by 2.6 km$^2$ (Figure 3b). Similarly, in 2013, AL led the spatial coverage of DC48 (Figure 3c). During this period, about 2597.1 km$^2$ (63.6%) of land was completely occupied by AL. This shows an intensification of 1846 km$^2$ from the 2003 coverage. The remaining land-use types were WeL, which covered 654.3 (16%); DF, 253.2 (6.2%); BL, 227.3 (5.6%); ShL, 97.7 (2.4%); MQ, 101.5 (2.5%); BuA, 59 (1.4%); GL, 34.2 (0.8%); and WB, 10.2 km$^2$ (0.2%). Likewise, in the year 2022, WeL covered an area of 1120.5 km$^2$ (27.4%). This depicts an increase of 466.3 km$^2$ from its earlier coverage (Figure 3d). Similarly, ShL covered 515.5 (12.6%), GL covered 157.8 (3.9%), BuA covered 93.3 (2.3%), and WB covered 16.6 km$^2$ (0.4%). Additionally, we observed a substantial reduction in AL, MQ, BL, and DF by 42.4, 1.1, 4.3, and 5.6%, respectively. In addition, AL, BL, ShL, BuA, and DF decreased annually from 1993 to 2003 by 120.6, 82, 25.1, 11.6, and 0.3 km$^2$, respectively (Supplementary Figure S4a). However, for MQ, WeL, GL, and WB, there was a notable increase of 103.3, 90.9, 38.4, and 6.3 km$^2$. As seen in Supplementary Figure S4b, MQ, GL, WeL, BuA, WB, and BL experienced annual losses of 98.5, 43.8, 42.9, 24.3, and 5.7 km$^2$ from 2003 to 2013, respectively. In the same period, an increase in AL, DF, ShL, and BL by 184.7, 23.8, 7.4, and 0.1 km$^2$ was observed. Conversely, in the years 2013–2022, most of the land uses changed at different rates (Supplementary Figure S4c). For instance, AL diminished by 86.6, BL by 10.4, MQ by 5.6, and DF by 2.3 km$^2$. Meanwhile, WeL, ShL, GL, BuA, and

WB intensified by 46.8, 41.8, 12.4, 3.4, and 0.6 km$^2$, respectively. In general, BL, BuA, AL, and MQ declined annually by 102.5, 36.1, 25, and 0.9 km$^2$, respectively. However, over the past three decades, WeL, ShL, DF, GL, and WB increased by 105.1, 23.8, 23.6, 7.7, and 1.4 km$^2$, respectively (Supplementary Figure S4d).

The Sedibend (DC 42) District of Gauteng Province, which covered about 14.8%, was dominated by AL, i.e., 813.6 km$^2$ (30.3%), in 1993. Following this, BuA was the second overriding land-use type of the district. In this period, BuA covered 638.4 (23.8%), BL covered 622.7 (23.2%), WeL covered 214.3 (8%), ShL covered 196.3 (7.3%), GL covered 147.4 (5.5%), WB covered 27.8 (1%), MQ covered 13.7 (0.5%), and DF covered 12.6 km$^2$ (0.5%). In 2003, most of the earlier land-use types changed sustainably from the 1993 coverage. For instance, WeL increased by 860.8, MQ by 283.3, GL by 204, WB by 43.5, and DF by 5.9 km$^2$. On the contrary, BuA declined by 518.1, AL by 403.6, BL by 321.8, and ShL by 154.2 km$^2$. Likewise, during the year 2013, AL became the overriding land-use type, while GL covered a small portion of the district. During this year, AL covered 1524.5, WeL covered 426 (15.9%), BL covered 338.5 (12.6%), DF covered 124.4 (4.6%), BuA covered 90.7 (3.4%), ShL covered 86 (3.2%), WB covered 44.3 (1.6%), MQ covered 34.1 (1.3%), and GL covered 18.4 km$^2$ (0.7%). However, in 2022, a notable reduction in AL, BL, MQ, and DF was observed by 509.5, 63.9, 23.2, and 19.7 km$^2$, respectively, from the 2013 coverage. In the same year, WeL, ShL, GL, BuA, and WB intensified by 282.4, 150.5, 146.5, 24, and 6.9 km$^2$, respectively. Moreover, the BuA, AL, BL, and ShL declined annually by 51.8, 40.4, 32.2, and 15.4 km$^2$, respectively, from 1993 to 2003 (Supplementary Figure S5a). On the other hand, DF, WB, GL, MQ, and WeL intensified annually by 0.6, 4.4, 20.4, 28.3, and 86.1 km$^2$, respectively. However, in the period 2003–2013, most of the land-use types changed. For instance, WeL lost 64.9, GL lost 33.3, MQ lost 26.3, BuA 3, and WB lost 2.7 km$^2$ each year (Supplementary Figure S5b). In the same period, BL, ShL, DF, and AL expanded by 3.8 km$^2$, 4.4 km$^2$, 10.6 km$^2$, 111.5 km$^2$, respectively (Supplementary Figure S5c). Moreover, a significant loss in AL, BL, MQ, and DF were observed annually by 50.9, 6.4, 2.3 km$^2$, and 1.4 km$^2$, respectively (Supplementary Figure S5c). Conversely, the crucial land-use types of WB, BuA, GL, ShL and WeL intensified annually by 0.7, 2.4, 14.6, 15, and 28.2 km$^2$, respectively. In general, BuA, BL, and MQ dropped annually by 58.2, 38.7, and 0.3 km$^2$, respectively. Those loss became a gain to other land-use types, namely as GL, WB, ShL, DF, AL, and WeL; in these land-use types, annual increases of 1.6, 2.6, 4.5, 10.9, 22.4, and 54.9 km$^2$ were observed between 1993 and 2022 (Supplementary Figure S5d).

### 3.3. Surface Water Bodies Analysis

Table 5 indicates the overall surface water bodies of the study area, using the MNDWI and NDWI. The results show that the surface water content has been below the standard and diminishing yearly over the last three decades (Figure 5a–h). This reduction may have caused the worsening shortages of surface water access for livestock and humans in all district municipalities of GP from 1993 to 2022 (Table 5). As a result, the incidence of drought would be severe [54], as well as the regularity, intensity, severity, and spatial extent. Although all district municipalities of GP are depicted as being below the standard thresholds, the TSH and EKU show relatively higher MNDWI values than all of the districts of GP. In these districts, the max MNDWI values during the year 1993 were 0.39, and 0.38, respectively. In the same year, the district of DC42 revealed a lower (0.27) MNDWI value. In addition, the NDWI shows very low surface water content levels compared to the MNDWI across all the study sites. Studies indicated that surface water bodies may be quickly affected by extreme weather and poor land-use management-related factors. Additionally, the allocation of water bodies using the RF method, MNDWI, and NDWI was analyzed. The findings showed that the spatial representations of water bodies in the province of Gauteng varied in three of the models. For instance, the MNDWI and NDWI show a decline of 103 and 594 km$^2$, respectively, from 1993 to 2022, whereas the RF model indicates an increase of 38.56 km$^2$.

**Table 5.** Comparative analysis of surface water bodies across each district municipality in Gauteng Province (GP).

| Year | Indices | DM | Min | Max | Mean | STD | CV% |
|------|---------|-----|------|------|-------|------|--------|
| 1993 | MNDWI | JHB | −0.48 | 0.32 | −0.19 | 0.06 | −30.11 |
| | | TSH | −0.75 | 0.39 | −0.22 | 0.05 | −21.54 |
| | | EKU | −0.61 | 0.38 | −0.21 | 0.06 | −28.54 |
| | | DC42 | −0.75 | 0.27 | −0.22 | 0.06 | −27.96 |
| | | DC48 | −0.73 | 0.34 | −0.24 | 0.04 | −17.77 |
| | NDWI | JHB | −0.46 | 0.20 | −0.14 | 0.04 | −31.79 |
| | | TSH | −0.50 | 0.22 | −0.15 | 0.03 | −21.15 |
| | | EKU | −0.46 | 0.22 | −0.13 | 0.04 | −29.48 |
| | | DC42 | −0.46 | 0.19 | −0.14 | 0.04 | −28.70 |
| | | DC48 | −0.47 | 0.26 | −0.15 | 0.03 | −20.80 |
| 2003 | MNDWI | JHB | −0.71 | 0.50 | −0.19 | 0.05 | −27.54 |
| | | TSH | −0.75 | 0.47 | −0.24 | 0.05 | −19.64 |
| | | EKU | −0.76 | 0.37 | −0.22 | 0.06 | −26.55 |
| | | DC42 | −0.63 | 0.48 | −0.23 | 0.06 | −27.07 |
| | | DC48 | −0.75 | 0.47 | −0.24 | 0.04 | −17.79 |
| | NDWI | JHB | −0.75 | 0.51 | −0.17 | 0.06 | −34.45 |
| | | TSH | −0.75 | 0.52 | −0.16 | 0.04 | −22.47 |
| | | EKU | −0.73 | 0.29 | −0.16 | 0.05 | −29.42 |
| | | DC42 | −0.74 | 0.52 | −0.14 | 0.04 | −28.89 |
| | | DC48 | −0.73 | 0.50 | −0.16 | 0.04 | −24.04 |
| 2013 | MNDWI | JHB | −0.98 | 0.35 | −0.18 | 0.06 | −31.44 |
| | | TSH | −0.87 | 0.41 | −0.24 | 0.05 | −22.33 |
| | | EKU | −0.92 | 0.40 | −0.22 | 0.06 | −28.93 |
| | | DC42 | −0.77 | 0.33 | −0.24 | 0.07 | −29.95 |
| | | DC48 | −0.83 | 0.43 | −0.26 | 0.05 | −17.44 |
| | NDWI | JHB | −0.96 | 0.35 | −0.15 | 0.05 | −32.36 |
| | | TSH | −0.80 | 0.34 | −0.16 | 0.04 | −22.54 |
| | | EKU | −0.89 | 0.28 | −0.15 | 0.05 | −31.11 |
| | | DC42 | −0.50 | 0.32 | −0.15 | 0.05 | −33.80 |
| | | DC48 | −0.76 | 0.32 | −0.17 | 0.04 | −21.79 |
| 2022 | MNDWI | JHB | −0.89 | 0.41 | −0.18 | 0.05 | −29.93 |
| | | TSH | −0.86 | 0.40 | −0.23 | 0.05 | −22.92 |
| | | EKU | −0.97 | 0.38 | −0.21 | 0.06 | −28.23 |
| | | DC42 | −0.77 | 0.34 | −0.23 | 0.07 | −29.16 |
| | | DC48 | −0.74 | 0.39 | −0.25 | 0.05 | −18.59 |
| | NDWI | JHB | −0.86 | 0.37 | −0.15 | 0.05 | −30.58 |
| | | TSH | −0.84 | 0.40 | −0.16 | 0.04 | −25.38 |
| | | EKU | −0.96 | 0.29 | −0.15 | 0.04 | −29.31 |
| | | DC42 | −0.48 | 0.22 | −0.16 | 0.05 | −32.87 |
| | | DC48 | −0.50 | 0.27 | −0.16 | 0.04 | −23.62 |

*3.4. Relationships of Various Surface Water Content Indicators at Various Radiuses*

In this study, we examined the statistical relationship between the MNDWI and NDWI at a 5, 10, and 15 km radius from the center of the study area to ascertain how the two significant surface water content indicators related to one another. Figure 6a,b depict the long-term mean NDWI and MNDWI levels at various radiuses.

Our results indicated that the Pearson correlation between the MNDWI and NDWI is higher at the 5 km radius than at 10 and 15 km (r = 0.60, $p = 0.00$, n = 87,260) and is statistically significant ($p = 0.000$) (Figure 7a–c).

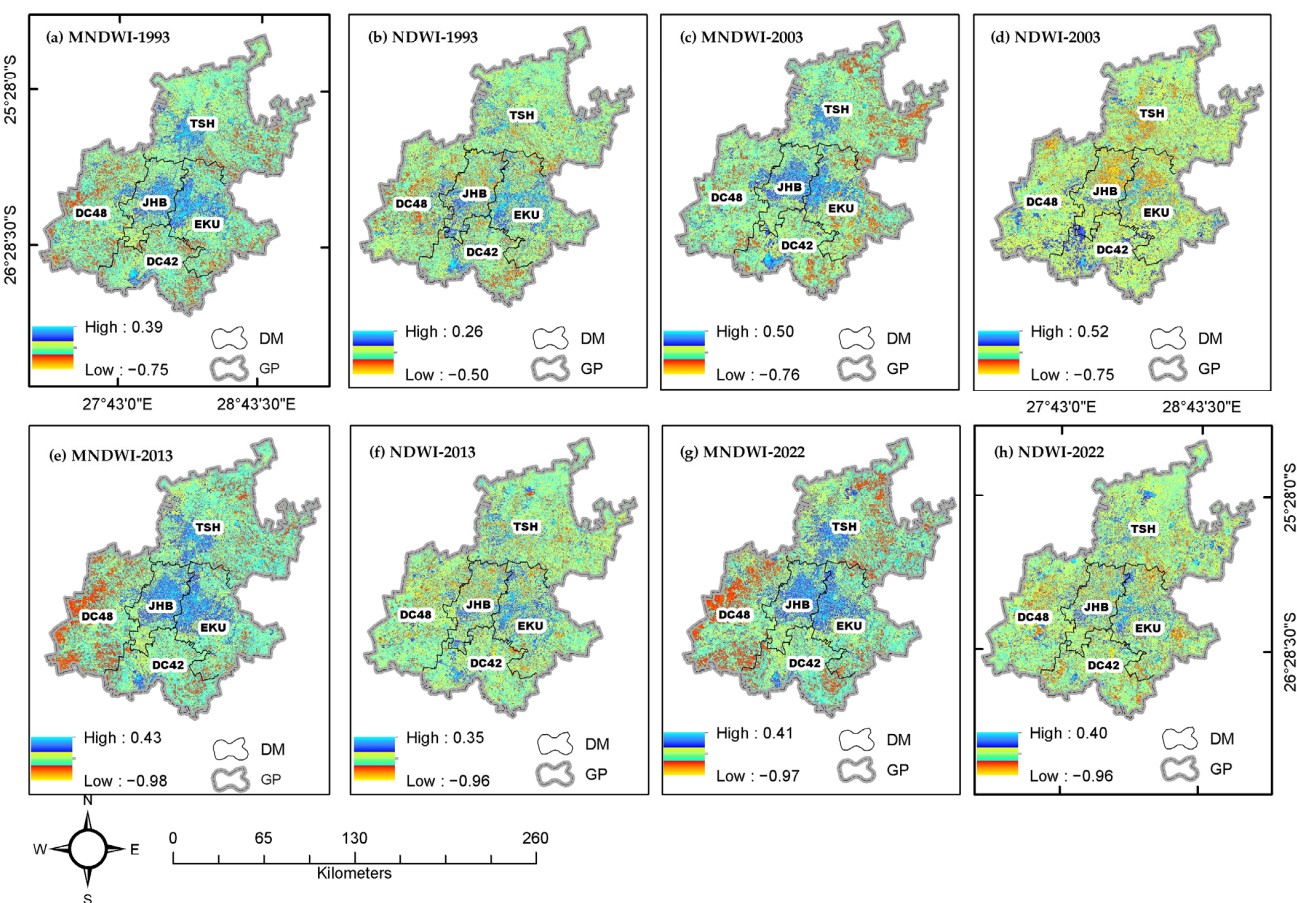

**Figure 5.** (**a**–**h**) Surface water content detection in Gauteng Province across each district municipality.

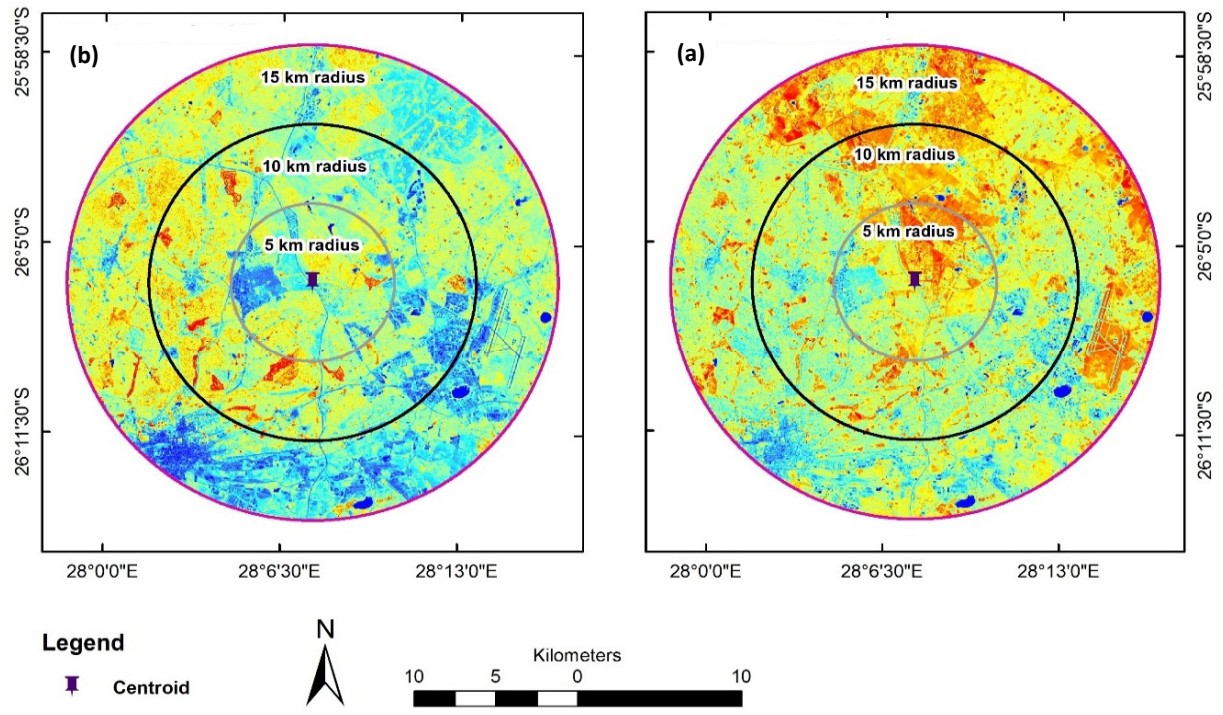

**Figure 6.** (**a**,**b**) The long-term mean NDWI (**a**) and MNDWI (**b**) levels at various radiuses from 1993–2022.

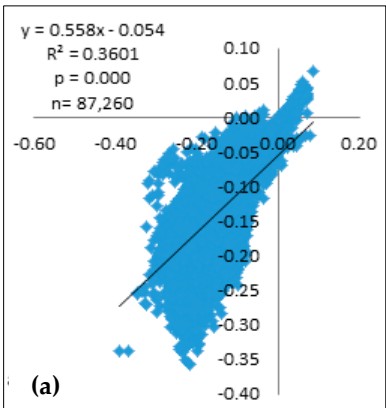
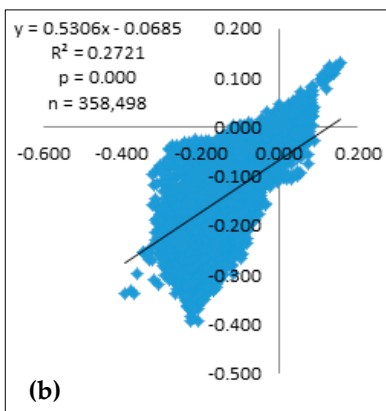
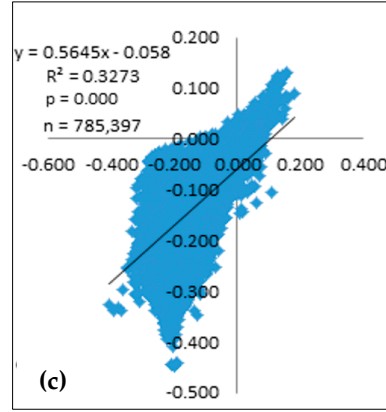

**Figure 7.** (**a**–**c**) Statistical relationships between the MNDWI and NDWI at 5 km (**a**), 10 km (**b**), and (**c**) 15 km radiuses from the center of the study area.

## 4. Conclusions

This study explored the performance of machine-learning models such as RF to characterize the impact of changing land-use diversity on surface water bodies. Our results showed that the spatial coverage of BL has decreased by 100.4 km$^2$ annually during the last three decades. The influence of the dense vegetation in GP was also evident in BuA, which showed a decreasing trend of 42.2 km$^2$. On the other hand, WB, MQ, AL, GL, ShL, DF, and WeL have increased by 1.3, 2.3, 2.9, 5.6, 11.2, 29.6, and 89.5 km$^2$ annually. Between 1993 and 2022, the spatial coverage of MQ, WB, GL, and DF increased in EKU by 0.1, 1.5, 6.2, and 10.8 km$^2$, respectively. During the same time period, the spatial extent of BL, AL, BuA, ShL, and WeL decreased by an average of 22.8, 11.2, 7.5, 3.7, and 1.9 km$^2$, respectively. In the metropolitan city of JHB, the coverage of ShL, which is crucial for minimizing the local effects of climate change, has typically decreased. In this city, ShL has declined by 23.4%, BL by 5.4%, AL and GL by 2.1, and WeL by 0.6 km$^2$ yearly. However, there were annual increases in WB, MQ, BuA, and DF of 0.3, 2.3, 3.2, and 5 km$^2$, respectively. Additionally, the spatial area of BL, AL, and BuA decreased by 142.8, 54.1, and 35 km$^2$ in the TSH District, respectively. However, ShL, WB, MQ, GL, DF, and WeL showed an annual improvement of 0.9, 2.3, 2.5, 5.1, 31.9, and 151.8 km$^2$. In DC48, BL, BuA, AL, and MQ declined by 102.5, 36.1, 25, and 0.9 km$^2$, respectively. Nonetheless, WeL, ShL, DF, GL, and WB have intensified by 71.8, 26.8, 23.6, 7.7, and 1.4 km$^2$, respectively. Additionally, BuA, BL, and WB decreased annually by 60.6, 49.9, and 1.2 km$^2$ in the DC42 District. Other land-use types, such as GL, MQ, DF, ShL, WeL, and AL, intensified annually by 1.2, 3.5, 24.2, 35.5, 67.4, and 101.9 km$^2$, respectively. Additionally, this study reported that the levels of surface water bodies are decreasing because of the severe effects of land-use change. This study found that the statistical relationship between the MNDWI and NDWI was higher at a radius of 5 km ($r = 0.60$, $p = 0.00$, $n = 87{,}260$) than at 10 and 15 km. The finding of this study helps us to comprehend the implications of the changing land-use diversity on surface water bodies and improves the existing land-use system. Additionally, it can be used as a baseline study for future research on the economic effects of land-use change on surface water resources.

**Supplementary Materials:** The following supporting information can be downloaded at: https://www.mdpi.com/article/10.3390/rs15164092/s1. All supplementary annexes, such as land-use diversity classes, trend at municipality levels, and RF–GEE scripts are placed here as supplementary information to enhance the quality of our manuscript.

**Author Contributions:** E.G. and P.M. conducted the research and published the manuscript. All authors have read and agreed to the published version of the manuscript.

**Funding:** This research was funded by the University of Witwatersrand, grant number E.G postdoctoral research fellowship 2023, and the APC was funded by the School of Geography, Archaeology, and Environmental Studies (GAES), University of Witwatersrand, Johannesburg, South Africa.

**Data Availability Statement:** Any datasets used in our study will be shared upon request to the corresponding author.

**Acknowledgments:** We are indebted to the financial support of the University of the Witwatersrand School of Geography, Archaeology, and Environmental Studies (GAES), Johannesburg, South Africa.

**Conflicts of Interest:** The authors declare no conflict of interest.

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
