# Peer review of "An Application of Machine-Learning Model for Analyzing the Impact of Land-Use Change on Surface Water Resources in Gauteng Province, South Africa"

_remotesensing, doi:10.3390/rs15164092_

Round 1

Reviewer 1 Report

Dear authors.

I am grateful for the opportunity to review the article "An application of machine learning model for analyzing the impact of land use change on surface water resources in Gauteng, South Africa". I would like to note its high scientific level and extreme relevance. I will not dwell on the merits of this article, they are already obvious. I will note only those shortcomings and questions that I found when I read the article. I believe that the article can be accepted after the authors' responses to my comments:

1. «Since the 1960s, the biophysical properties of the land surface have changed by an average value of 720,000-kilometer square (km2) each year». Probably, you mean – what is it all over the world? Or in Africa? Clarify this. This applies to the entire section 1. You provide statistics, but do not specify which region of the world this applies to.

2. Check the correctness of the links (For example, "For instance, Yang & Huang (2021) reported ..."). Check again the correctness of the article according to the rules of the scientific journal.

3. In Figure 1, check the correctness of the jackals heights. You probably have a mistake because the study area contrasts along the borders with the heights of the adjacent territories. Obviously, the elevation map of South Africa should have its own legend of absolute heights, a coordinate grid. Sign the neighboring countries, seas and oceans.

4. The research methodology is written extremely concisely and unclear. If you used GEE, then specify in the additional materials to the article the code that you used. Is the description you provided typical for any GIS research? What programs have you used? This section is the weakest. It needs to be radically redone.

5. What does "Total" mean in Figures 4-9. Did I not see this in the drawings themselves?

6. Figure 10 shows the legends of maps that have different borders. It is probably necessary to make a single legend of each of the indexes. Otherwise, it misleads the reader.

7. Figure 12. Are there any typos in the meaning of "p"?

8. What is RF? random forest? I didn't see the abbreviation decoding. Why are you highlighting water bodies without using RF? What are the differences between the allocation of water bodies by RF, MNDWI and NDWI?

9. The section "Results and Discussion" contains only the results of the study. Unfortunately, I didn't see exactly "Discussions" in this section. I did not see the limitations of research, comparison of the results with other regions of the world, and generally no discussion of the results.

10. Section 4 is a simple enumeration of statistically obtained data. It is necessary to expand it, and significantly reduce the number of statistics /figures. Indicate the prospects for further research.

Author Response

Response to Reviewer 1 Comments

Point 1:  «Since the 1960s, the biophysical properties of the land surface have changed by an average value of 720,000-kilometer square (km2) each year». Probably, you mean – what is it all over the world? Or in Africa? Clarify this. This applies to the entire section 1. You provide statistics, but do not specify which region of the world this applies to.

Response 1: Dear reviewer, Since the 1960s, the biophysical features of the earth surface have changed by an average value of 720,000-kilometer square (km2) globally each year [1]. The changes have occurred globally not only in Africa or some specific parts of the globe. This statement is also supported by literature in our revised manuscript.

Point 2:  Check the correctness of the links (For example, "For instance, Yang & Huang (2021) reported ..."). Check again the correctness of the article according to the rules of the scientific journal.

Response 2: Dear Reviewer, your constructive feedback has been included in our revised manuscript.

Point 3:  In Figure 1, check the correctness of the jackals heights. You probably have a mistake because the study area contrasts along the borders with the heights of the adjacent territories. Obviously, the elevation map of South Africa should have its own legend of absolute heights, a coordinate grid. Sign the neighboring countries, seas and oceans.

Response 3: Dear reviewer, we enhanced Figure 1 in response to your helpful suggestions. Thank you.

Point 4:  The research methodology is written extremely concisely and unclear. If you used GEE, then specify in the additional materials to the article the code that you used. Is the description you provided typical for any GIS research? What programs have you used? This section is the weakest. It needs to be radically redone..

Response 4: Dear reviewer, yes, the RF machine learning model in the GEE platform was used to analyze all our remote sensing data sets. The source code or java script for GEE is provided as supplementary information. All GIS research can use the description, but the objectives must be the same. We used RF as a program on the GEE platform.

Point 5:  What does "Total" mean in Figures 4-9. Did I not see this in the drawings themselves?

Response 5: Dear reviewer, the word "total" was automatically formed when we selected the waterfall chart type. Now, we have removed it from our graphs because it does give sense on the meaning of the increasing and decreasing trends of land use diversity.

Point 6:  Figure 10 shows the legends of maps that have different borders. It is probably necessary to make a single legend of each of the indexes. Otherwise, it misleads the reader.

Response 6: Dear reviewer, we tried to have a single legend for each index, but the problem is that the highest and lowest or max and min values of each year are different. Therefore, we cannot have a single legend that can commonly represent the max and min values of each index for the period 1993, 2003, 2013, and 2022.

Point 7:  Figure 12. Are there any typos in the meaning of "p"?

Response 7: Dear reviewer, yes, there were typos in Figure 12-c. In the updated version of the manuscript, the mistake has been fixed.

Point 8:  What is RF? random forest? I didn't see the abbreviation decoding. Why are you highlighting water bodies without using RF? What are the differences between the allocation of water bodies by RF, MNDWI and NDWI?

Response 8: The term RF, which stands for random forest, is used frequently in this text. Without RF, we did not identify any land use types. RF models in the GEE platform were used to analyze all sorts of land use. The Java script or code for GEE is included with this document as supplemental information. The research area's lake, which is depicted in blue in Figure 1, is only there for description. Our revised work includes the differences in the distribution of water bodies by RF, MNDWI, and NDWI.

Point 9:  The section "Results and Discussion" contains only the results of the study. Unfortunately, I didn't see exactly "Discussions" in this section. I did not see the limitations of research, comparison of the results with other regions of the world, and generally no discussion of the results.

Response 9: Dear reviewer, we improved the discussion section of the manuscript. Our findings were triangulated with those of other studies. The limitations of our study are also listed as potential areas for further research in the manuscript's conclusion section.

Point 10:  Section 4 is a simple enumeration of statistically obtained data. It is necessary to expand it, and significantly reduce the number of statistics /figures. Indicate the prospects for further research.

Response 10: Dear reviewer, as our land use types are nine, it is difficult for us to ignore the potential land use types to reduce the number of statistics due to the nature of the study. We analyzed different land use diversity types at province and municipality level during the last four decades. However, we clearly indicated the prospective for further research in the conclusion section of the manuscript.

Thank you very much for your constructive comments.

Best regards,

Reviewer 2 Report

good paper with clear contributions, this paper explored the implications of  changing land use diversity on surface water resources using a random forest classifier machine learning algorithm and remote sensing models in Gauteng province, South Africa.

Author Response

Response to Reviewer 2 Comments

General feedback: The paper ‘An application of machine learning model for analyzing the impact of land use change on surface water resources in Gauteng, South Africa’ provides an interesting and innovative study. I do not have any major revisions for the paper, so my recommendation was for 'Accept after minor revision'.

Response: Dear reviewer, thank you very much for your feedback.

Point 1:  Use km² and not km2. Corrections on lines 228, 242 and Figures 4 and 5. Proofread all text.

Response 1: Dear reviewer, we improved our manuscript as per your constructive feedback..

Point 2:  Correct spacing in the text, they are not in the same pattern. On line 229 use 27º 17' 15" and not 27º17' 15" for example.

Response 2: Dear reviewer, we corrected the space and patterns.

Point 3:  Missing parentheses. Line 239.

Response 3: Dear reviewer, many thanks for the comments. The missing parentheses were added to our updated manuscript.

Point 4:  Correct the paragraph and make a reference where you must indicate the site and date of access. Line 265.

Response 4: Dear reviewer, in response to the helpful suggestions, we improved the paragraph (phrase).

Point 5:  Use dots and not commas to separate decimals. Check Figure 4.

Response 5: Dear reviewer, we applied dots to separate decimals throughout the figures.

We greatly appreciate your constructive criticism.

Best regards,

Reviewer 3 Report

The paper ‘An application of machine learning model for analyzing the impact of land use change on surface water resources in Gauteng, South Africa’ provides an interesting and innovative study. I do not have any major revisions for the paper, so my recommendation was for 'Accept after minor revision'.

Additional comments

1 - Use km² and not km2. Corrections on lines 228, 242 and Figures 4 and 5. Proofread all text.

2 - Correct spacing in the text, they are not in the same pattern. On line 229 use 27º 17' 15" and not 27º17' 15" for example.

3 - Missing parentheses. Line 239.

4 - Correct the paragraph and make a reference where you must indicate the site and date of access. Line 265.

5 - Use dots and not commas to separate decimals. Check Figure 4.

Author Response

General feedback: The paper ‘An application of machine learning model for analyzing the impact of land use change on surface water resources in Gauteng, South Africa’ provides an interesting and innovative study. I do not have any major revisions for the paper, so my recommendation was for 'Accept after minor revision'.
Response: Dear reviewer, thank you very much for your feedback.
Point 1: Use km² and not km2. Corrections on lines 228, 242 and Figures 4 and 5. Proofread all text.
Response 1: Dear reviewer, we improved our manuscript as per your constructive feedback..
Point 2: Correct spacing in the text, they are not in the same pattern. On line 229 use 27º 17' 15" and not 27º17' 15" for example.
Response 2: Dear reviewer, we corrected the space and patterns.
Point 3: Missing parentheses. Line 239.
Response 3: Dear reviewer, many thanks for the comments. The missing parentheses were added to our updated manuscript.
Point 4: Correct the paragraph and make a reference where you must indicate the site and date of access. Line 265.
Response 4: Dear reviewer, in response to the helpful suggestions, we improved the paragraph (phrase).
Point 5: Use dots and not commas to separate decimals. Check Figure 4.
Response 5: Dear reviewer, we applied dots to separate decimals throughout the figures.
We greatly appreciate your constructive criticism.

Best regards,

Reviewer 4 Report

Please see attached file. It includes typed and handwritten comments. The handwritten comments stop at page 17, so you are not missing anything.

Minor English language edits suggested within the attached comments.

Author Response

Response to Reviewer 4 Comments

General feedback: This paper researches an application of machine learning for measuring land use change over time in South Africa. The premise is strong, and the abstract is very straight forward and informative, however, I feel that a lot of work needs to be done on the format of the paper, especially staring with the results section. I feel that there are entirely too many abbreviations/acronyms in this paper, which makes it hard to follow along. The authors also need to work on the management of white space in the paper. The paper could be about 4 pages shorter with properly managed with space. Also, since the paper stops using line numbers on pages 13, I have attached handwritten notes to go along with my comments.

Response: Dear reviewer, thank you very much for your feedback. Your constructive feedback is well articulated in the revised version of our manuscript. Kindly see the revised manuscript. Thank you.

Point 1:  Abstract Line 14: I suggest changing “depleted alarmingly” to depleted at an alarming rate.

Response 1: Dear reviewer, your constructive feedback has been included in our revised manuscript.

Point 2: Line 25 “2” should be superscript before “respectively.”

Response 2: Dear reviewer, your constructive feedback has been included in our revised manuscript.

Point 3:  Line 30 suggest changing to “ to reduce the environmental impacts of land use change.”

Response 3: Dear reviewer, your constructive feedback has been included in our revised manuscript.

Point 4:  Line 31: I suggest removing keywords that are already in the little (e.g., changing-learning, Gauteng, South Africa). Since these words are already in the title, they will show up in literature searches. The keyword section is a chance for you to add additional search terms, so you should choose terms that aren’t already being used.

Response 4: Dear reviewer, your constructive feedback has been included in our revised manuscript.

Point 5:  Line 35: what “biophysical properties?” please reword this to be more explicit.

Response 5: Dear reviewer, the terms have been changed to biophysical features to make it clearer.

Point 6:  Line 39: suggest changing to “land surface processes, diversity, and change requires.”

Response 6: Dear reviewer. your constructive feedback has been included in our revised manuscript..

Point 7:  Line 47: When directly referencing a specific paper (or when starting a sentence with a citation), it is best to state their name in this and then put the bracketed citation following. In order words, don’t use [1] as a noun, use the actual noun and then cite it. So, in this case, the sentence would read: “For instance, Winkler et al. [1] reported the loss.”

Response 7: Dear reviewer. your constructive feedback has been included in our revised manuscript.

Point 8:  Lines 60-63: These sentences seem to contradict each other. First sentence says that there has been a loss of cropland and water. Second sentence says that cropland has risen by 110%. I think that you mean that loss of cropland is one area has caused the loss of forest in other areas. If so, this needs to be made clearer. If not, then something needs to be made with these two sentences, b/c of the contradiction.

Response 8: Dear reviewer, the statement is paraphrased into “Additionally, the spatial extent of forest land has decreased by 17% [7].”

Point 9:  Line 65 suggest changing to “degradation resulting in loss” (rather than to loss)

Response 9: Dear reviewer, your constructive feedback has been included in our revised manuscript.

Point 10:  Line 74 “how quickly changes the land and ameliorate the existing land use planning” is confusing. I suggest rewording this sentence. May apologies for not have a suggestion, because I am not exactly sure what the original intent is.

Response 10: Dear reviewer, the statement is paraphrased into” Understanding the spatial patterns and processes will make it easier to predict where and how quickly the land will change and improve the existing land use planning [11]”.

Point 11:  Line 81: Again, spell out a direct reference to author “ For instance, Wulder et al. [14]”

Response 11: Dear reviewer. your constructive feedback has been included in our revised manuscript.

Point 12:  Line 89 : Again, spell out a direct reference to author “ For instance, Wulder et al. [14]”

Response 12: Dear reviewer. your constructive feedback has been included in our revised manuscript.

Point 13:  Change quantifying to quantify.

Response 13: Dear reviewer. your constructive feedback has been included in our revised manuscript.

Point 14:  Line 101 Use the name again in the citation here

Response 14: Dear reviewer. your constructive feedback has been included in our revised manuscript.

Point 15:  Lines 108-109 I am pretty sure that South Africa is bigger than 1.2 km2. Perhaps you were referencing a specific land cover type?

Response 15: Dear reviewer. It is corrected now. It was typo error. The total land mass of South Africa is about 1,221,037 (one million, two hundred twenty-one thousand and thirty-seven) km2.

Point 16:  Line 132 I suggest adding a “a” between “using” and “machine-learning approach”

Response 16: Dear reviewer, your constructive feedback has been included in our revised manuscript.

Point 17:  Line 135 I suggest adding “more” before “robust.

Response 17: Dear reviewer, your constructive feedback has been included in our revised manuscript.

Point 18:  Line 152 Suggest changing to “ArcGIS software, and the Modified Normalized Difference Water Index.”

Response 18: Dear reviewer, your constructive feedback has been included in our revised manuscript.

Point 19:  Line 166 Again, start the sentence with the author’s name and then add the citation.

Response 19: Dear reviewer, your constructive feedback has been included in our revised manuscript.

Point 20:  Line 189 Same as previous comment.

Response 20: Dear reviewer, your constructive feedback has been included in our revised manuscript.

Point 21:  Line 200 place “and” before “ii)”.

Response 21: Dear reviewer, your constructive feedback has been included in our revised manuscript.

Point 22:  Line 206 do not use contractions, spell out “were not”

Response 22: Dear reviewer, it has been corrected now.

Point 23:  Line 208 suggest changing “municipality” to “municipalities.”

Response 23: Dear reviewer, it is corrected now.

Point 24:  Line 213 insert “and” before “iv)”

Response 24: Dear reviewer, it has been corrected now.

Point 25:  Lines 236-238 The reported average temperature is lower than the reported minimum temperature.

Response 25: Dear reviewer, it has been corrected now.

Point 26:  Line 268 The word “besides” appears as a sentence starter 7 times in this paper. I would suggest replacing it throughout with other words such as: Additionally, Furthermore, Moreover, In addition. Maybe it’s just me, but “besides,” sounds too contradictory, and the way it is used throughout the paper, seems to me that you need a term that is more complimentary.

Response 26: Dear reviewer, your constructive feedback has been included in our revised manuscript.

Point 27:  Line 272 add name to citation 35 since it is a direct reference.

Response 27: Dear reviewer, we added the name of the author as per the feedback.

Point 28:  Line 305 add name to citation 21 since it is a direct reference.

Response 28: Dear reviewer, we added the name of the author.

Point 29:  Line 311 This goes for all the equations; they need to be centered and the equation numbers need to all line up.

Response 29: Dear reviewer, your constructive feedback has been included in our revised manuscript.

Point 30:  Line 311 This goes for all the equations; they need to be centered and the equation numbers need to all line up.

Response 30: Dear reviewer, It has now been fixed.

Point 31:  Lines 313-317 This paragraph is double spaced instead of single spaced.

Response 31: Dear reviewer, It has now been fixed.

Point 32:  Lines 343-345 double spaced again.

Response 32: Dear reviewer, your constructive feedback has been included in our revised manuscript.

Point 33:  Lines 345 It is common to have the “o” underlined in the abbreviation for number?

Response 33: Dear reviewer, Yes it is. The typical abbreviation for the word number is “No.” But we can also write as “No”. However, we removed the underline “o” in our revised manuscript.

Point 34:  Lines 348 need to manage whitespace better, there are about 4 wasted pages due to whitespace.

Response 34: Dear reviewer, We were able to eliminate the whitespace.

Point 35:  Lines 393 use authors name on citation 47

Response 35: Dear reviewer, it is corrected now.

Point 36:  Lines 401 center the equation.

Response 36: Dear reviewer, we adjusted the alignment of all our equations into center.

Point 37:  Figure 2 don’t wrap (hyphenate) text in the diamond. Put “result” on one line, and all of “evaluation” on the second line. Do the same for “Quantitative Analysis.”

Response 37: Dear reviewer, Figure 2 shows the overall approaches that we have applied to analyze our data sets obtained from Landsat. The text in the diamond indicates (e.g., results evaluation) the procedure only how we conducted the research.  Here, our results are enormous. As a result, we cannot display a single result in the diamond because the results for 1993, 2003, 2013, and 2022 are all different.

Point 38:  Results and Discussion: Line numbers are gone. Below are some vague comments based upon page numbers, see attachment for handwritten notes.

Response 38: Dear reviewer, your constructive feedback has been included in our revised manuscript.

Point 39:  section 3.1 I assume “k-coefficient” is the kappa statistic. But later in the same paragraph, you call it the kappa index of agreement. If these are truly the same, then it is best to just choose one name and stick with it.

Response 39: Dear reviewer, it is corrected now.

Point 40:  Bottom of page 13 I can’t tell if the abbreviations here are supposed to be their own paragraph, or if it was meant to be part of the caption for Figure 3. If it goes with figure 3, then the font needs to be adjusted. If it is its own paragraph, then it needs to be read like a complete sentence.

Response 40: Dear reviewer, your constructive feedback has been included in our revised manuscript.

Point 41:  Top of page 15 Use the authors name here for [34]

Response 41: Dear reviewer, it is corrected now.

Point 42:  Figure 4 The text in the figure is fuzzy and hard to read. Need to use a higher resolution image.

Response 42: Dear reviewer, your constructive feedback has been included in our revised manuscript.

Point 43:  Too many figures I think that some of these figures could go into a supplement.

Response 43: Dear reviewer, our analysis was too detailed and that is why we have more figures. Now, we have reduced the figures and put them under the supplementary. Kindly see our revised manuscript.

We greatly appreciate your constructive criticism.

Best regards,

Round 2

Reviewer 1 Report

Dear Authors and Editor,

I am satisfied with the responses provided to almost all of my inquiries. However, I would like to make the following recommendations to the authors:

1. Please ensure the accuracy of figure numbering. There are two instances of Figure 4 in the article, followed by Figure 11.

2. In lines 63-64, I suggest either creating a unified legend or formatting Figures a-d individually. Since these figures are visually adjacent, readers are naturally inclined to compare them.

3. Referring to lines 266-268, this concern was raised earlier as well. If combining legends for NDVI and MNDWI is not feasible, I propose utilizing GIS-based classification. Readers often compare colors first, and having different quantitative values in the legend for the same color might lead to confusion.

Author Response

Response to Reviewer 1 Comments

General feedback: I am satisfied with the responses provided to almost all of my inquiries. However, I would like to make the following recommendations to the authors:

Response: Dear reviewer, thank you very much for the general feedback.

Point 1: Please ensure the accuracy of figure numbering. There are two instances of Figure 4 in the article, followed by Figure 11.

Response 1: Dear reviewer, thank you very much for the critical feedback. Yes, we observed that there were two instances of figure 4. It is corrected now in our revised manuscript.

Point 2: In lines 63-64, I suggest either creating a unified legend or formatting Figures a-d individually. Since these figures are visually adjacent, readers are naturally inclined to compare them.

Response 2: Dear reviewer, this figure gives comprehensive information regarding the trends of land use diversity classes transfer out and in across all the study years. Therefore, the existing legend and captions, such as a-, b-, c-, and d of each figure help readers to clearly understand the trends among the different years. If we put them separately into four different figures then our readers might be challenged in comparing about how much area has been gained and lost or which land use diversity class intensifying and shrinking during the study periods.

Point 3: Referring to lines 266-268, this concern was raised earlier as well. If combining legends for NDVI and MNDWI is not feasible, I propose utilizing GIS-based classification. Readers often compare colors first, and having different quantitative values in the legend for the same color might lead to confusion.

Response 3: Dear reviewer, we do understand your concern. Both of our figures are showing the amount or status of water contents in the same site at different radius using different remote sensing water content indices such as NDWI and MNDWI. Here, our main intention is to depict surface water content (bodies) at various radius from the centroid (i.e., the central part of the study area). Therefore, it is a must to have an identical color to depict the spatial extent of water bodies using the two different indices. In this case, the existing figure clearly shows the distribution of water bodies in the same area using different indicators and at different radiuses. The comparison is also across the indices and radius. However, if you are still interested and advising us to have a legend for each radius and index then we can create 6 different layouts (maps) to show the max and min surface water contents in the study area. Thank you so much for your constructive feedback.

Best regards,

Reviewer 4 Report

Thank you for the edits. I have just a few minor editing comments. When listing a long line of values in a single sentence (like lines 25, 104, and others throughout the results section), I think that it is okay to just list the units (km2 in this case) just once at the end, but I will leave that up to the editors.

The first mention of random forests is on line 136 (page 3), but you only use the abbreviation. You have random forest spelled out later with the abbreviation in parentheses on line 168. You need to flip that to where random forest is spelled out on line 136, and then the abbreviation used from there on out.

I have no other comments. Thank you for your timely edits! The paper has been greatly improved over the original version.

Cheers

Author Response

Response to Reviewer 4 Comments

Point 1: Thank you for the edits. I have just a few minor editing comments. When listing a long line of values in a single sentence (like lines 25, 104, and others throughout the results section), I think that it is okay to just list the units (km2 in this case) just once at the end, but I will leave that up to the editors.

Response 1: Dear reviewer, your constructive feedback has been included in our revised manuscript. Kindly see our revised manuscript. Thank you.

Point 2: The first mention of random forests is on line 136 (page 3), but you only use the abbreviation. You have random forest spelled out later with the abbreviation in parentheses on line 168. You need to flip that to where random forest is spelled out on line 136, and then the abbreviation used from there on out.

Response 2: Dear reviewer, your constructive feedback has been included in our revised manuscript. Kindly see our revised manuscript. Thank you.

Point 3: I have no other comments. Thank you for your timely edits! The paper has been greatly improved over the original version.

Response 3: Dear reviewer, your constructive feedback has been included in our revised manuscript. Kindly see our revised manuscript. Thank you.

Best regards,
